# DISTILLING REINFORCEMENT LEARNING INTO SINGLE-BATCH DATASETS

## ABSTRACT

Dataset distillation compresses a large dataset into a small synthetic dataset such that learning on the synthetic dataset approximates learning on the large dataset. Training on the distilled dataset can be performed in as little as one step of gradient descent. We demonstrate that distillation is generalizable to different tasks by distilling reinforcement learning environments into one-batch supervised learning datasets. This demonstrates not only distillation's ability to compress a reinforcement learning task but also its ability to transform one learning modality (reinforcement learning) into another (supervised learning). We present a novel extension of proximal policy optimization for meta-learning and use it in distillation of a multi-dimensional extension of the classic cart-pole problem, all MuJoCo environments, and several Atari games. We demonstrate distillation's ability to compress complex RL environments into one-step supervised learning, explore RL distillation's generalizability across learner architectures, and demonstrate distilling an environment into the smallest-possible synthetic dataset.

## 1 INTRODUCTION

As deep learning increases in popularity and is used on larger learning tasks, training costs impose a greater cost to the deep learning community and the planet. Methods to reduce training costs are more necessary than ever. We propose a method that can reduce reinforcement learning to a single step of gradient descent using an extension of dataset distillation.

Dataset distillation[1] is a technique in which a large dataset is compressed into a synthetic dataset, learnable in as few as a single step of gradient descent (Wang et al., 2018). The performance of a model when trained on the synthetic dataset should approximate the performance of that model when trained on the original dataset. By compressing the original dataset, distillation greatly reduces the cost of training models; in this work, we demonstrate one-step learning made possible with distillation. The compression provided by distillation can be used for low-resource training, interpretability, and data anonymization (Dong et al., 2022). The reduction of training costs resulting from this compression can accelerate searches such as hyperparameter searches, neural architecture searches, and high-performance initialization searches, and can produce inexpensive ensembles.

We propose a generalization of dataset distillation, which we call *task distillation*. Using techniques similar to dataset distillation, we posit that any learning task can be distilled into a compressed synthetic task. Task distillation allows for transmodal distillations, reducing a more complex learning task, such as a reinforcement learning environment, to a simpler learning task, such as a supervised classification dataset.

In order to advance task distillation beyond supervised-to-supervised distillation, we explore distilling reinforcement learning environments into supervised learning datasets, which we call *RL-to-SL distillation*. We choose to produce synthetic supervised datasets because supervised learning is the bread-and-butter of machine learning, with decades of research going into improving SL. We consider transforming an arbitrary learning task into supervised learning to be an important step to simplify machine learning. We select RL environments to distill because reinforcement learning has the added complexity of requiring exploration of the environment. Exploration can cause a host of

---

[1]Dataset distillation is related to model/policy distillation, but compresses the data rather than a trained model. Here, we utilize "distillation" to refer only to dataset distillation and its generalization, task distillation.

difficulties in learning, such as poor policy spaces that an agent cannot escape and a general increase in training complexity. Learning a successful policy often requires a great deal of exploration, which may involve learning on many repetitive observations that hold little useful information. By producing a compressed synthetic dataset, agents can learn an RL task quickly without exploration costs, as exploration has already occurred during distillation.

The paper begins by exploring distillation on the cart-pole problem (Barto et al., 1983) as a pedagogical exercise; then demonstrates distillation of more difficult RL environments [MuJoCo (Todorov et al., 2012), *Centipede* (Atari, 1980), *Ms. Pac-Man* (General Computer Corporation, 1982), *Pong* (Atari, 1972), and *Space Invaders* (Taito, 1978)]; and makes the following contributions:

1. Proposes a new formulation of RL task distillation using proximal policy optimization.

2. Proposes an N-dimensional extension of the cart-pole environment to allow for scaling the difficulty of cart-pole.

3. Demonstrates $k$-shot learning on single-batch datasets distilled from ND cart-pole using various initialization distributions and architectures; demonstrating distillation's generalization to unseen architectures.

4. Empirically validates the theoretical work of Sucholutsky & Schonlau (2021a) by demonstrating the minimum distillation sizes of environments with different numbers of action classes.

5. Provides a method that can be used to scale the difficulty of distillation for complex tasks.

6. Demonstrates distillation of complex Atari and the continuous MuJoCo environments.

## 2 BACKGROUND

### 2.1 DATASET DISTILLATION

Dataset distillation was originally proposed by Wang et al. (2018). This method utilizes meta-gradients to learn a synthetic dataset, based on how well a predictive model trained on the synthetic data performs on the true dataset. This process is split into two nested learning loops: in inner learning, a newly-initialized predictive model is trained on the synthetic dataset. In outer learning, the trained predictive model is tested against the true dataset, and the loss is backpropagated through the inner-learning process to the synthetic dataset. The synthetic dataset is updated with gradient descent, and the updated synthetic dataset is used to train a new predictive model in the next iteration.

The goal of dataset distillation is to produce a synthetic dataset $\{X_d, Y_d\}_\theta$ such that predictive models sampled from a given distribution $\lambda_\phi \in \Lambda$ can train on the synthetic dataset to reach high performance on a targeted training task $T_0$. The original formulation defines a fixed synthetic label vector $Y_d$; however, in this work we use the soft label formulation of Sucholutsky & Schonlau (2021b), in which $Y_d$ is a parameterized vector of real numbers that can be learned along with the synthetic data instances. The sampled learner is trained on the synthetic dataset using an appropriate loss measure; we utilize mean-squared error between the prediction and soft labels. This inner optimization must be differentiable, as outer learning requires backpropagating through the inner optimization. The trained model can be tested against a sample of $T_0$ with an appropriate loss metric (generally one that would be used in learning $T_0$ directly) to produce the outer meta-loss.

It is expected that the distillation's performance is on average less than or equal to the performance of the average model trained directly on the task. This is an expected trade-off for high compression and low training time, as the compressed dataset cannot contain all the information found in the full task. However, closely approaching this performance is vital for distillation's use cases.

We visualize a task-agnostic version of this formulation of dataset distillation in Figure 1.

### 2.2 RELATED WORKS IN DATASET DISTILLATION

A similar method to dataset distillation was proposed contemporaneously by Such et al. (2020). This method uses meta-gradient to train a generator model that produces the synthetic dataset, rather than directly learning the dataset. This method, the generative teaching network, was tested on

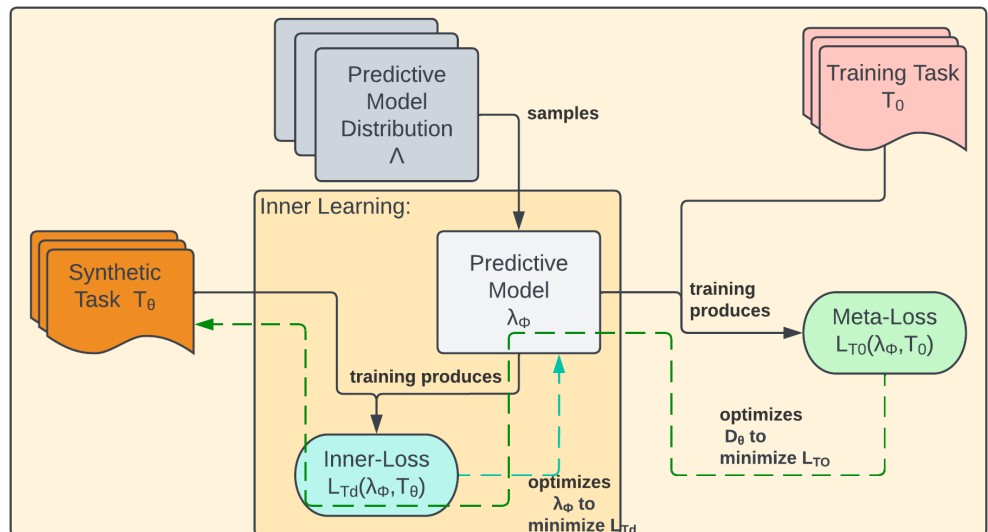

Figure 1: Training process for task-agnostic distillation based on Wang et al. (2018)

supervised learning tasks and on the simple RL cart-pole environment using A2C. Our experiments are inspired by this work, though we base our method on that of Wang et al. (2018) as our preliminary experiments indicated that parameterizing the dataset provided better results.

Other distillation methods exist, such as gradient matching (Zhao et al., 2021), trajectory matching (Cazenavette et al., 2022), distribution matching (Zhao & Bilen, 2023), differentiable Siamese augmentation (Zhao & Bilen, 2021), and kernel ridge-regression (Nguyen et al., 2021). We base our formulation of that of Wang et al. (2018) because of its simplicity and its independence from expert models and trajectories, though we posit that our method could be adapted to any of these formulations. Other non-distillation methods for simplifying learning, such as behavioral cloning, also require expert examples, and unlike distillation cannot feasibly be used to produce many models due to high training costs.

Lupu et al. (2024) has similar motivations to our work and also tackles distilling reinforcement learning environments. Their approach uses evolutionary strategies to optimize the dataset, while we utilize meta-gradients calculated with PPO loss. Our work distinguishes itself by focusing on minimum-sized datasets and single-step learning, which is feasible given the direction provided by the meta-gradients.

## 2.3 PROXIMAL POLICY OPTIMIZATION

PPO is a reinforcement learning algorithm that combines two policy gradient methods: actor-critic and trust region optimization. PPO uses two networks: the actor $\pi_\phi$ to learn the optimal policy, and the critic $V_\psi$ to approximate the value function (the return at a given state) given policy $\pi_\phi$. The actor determines the actions taken at each timestep at training and evaluation time, while the critic evaluates the actions taken during training only; it performs no function at evaluation time. PPO actor loss over a batch of a given trajectory of observations is as follows (Schulman et al., 2017):

$$L_\pi = -\frac{1}{b} \sum_1^n \min \left[ \frac{\pi_\phi^i(a_t|s_t)}{\pi_\phi^0(a_t|s_t)} A_t, \text{clip} \left( \frac{\pi_\phi^i(a_t|s_t)}{\pi_\phi^0(a_t|s_t)}, 1-\epsilon, 1+\epsilon \right) A_t \right] \tag{1}$$

where $b$ is the batch size, $\pi_\phi^0$ is the policy network with the parameters used to gather the observation sequences from the environment, $\pi_\phi^i$ is the policy network with its current parameters, $a_t$ is the action taken at time $t$, $s_t$ is the state at time $t$, $A_t$ is the advantage at time $t$, and $\epsilon$ is the trust region

width. $A_t$ can be calculated in various ways, but we use generalized advantage estimation in our implementation (Schulman et al., 2016).

PPO requires an auxiliary critic network, trained alongside the actor, to determine policy loss. The critic is not used at evaluation time and does not affect the policy after training concludes. The critic is trained to approximate the return $R_t$ expected from timestep $t$, given the agent is at state $s_t$ and is following the policy $\pi_\phi$. Batched critic loss is as follows (Schulman et al., 2017):

$$L_V = \frac{1}{n} \sum_1^n (V_\psi(s_t) - R_t)^2 \tag{2}$$

where $n$ is batch size, $V_\psi$ is the critic network, $s_t$ is the state at time $t$, and $R_t$ is the return at time $t$.

## 3 RL-TO-SL DISTILLATION

The learner, or distillation-trained predictive model, must be capable of performing on both the true task $T_0$ (i.e. act as an RL agent) and the distilled task $T_d$ (i.e. act as a classifier). In RL-to-SL distillation, a learner $\lambda$ must be capable of both producing a policy $\hat{p}$ for acting in the outer RL environment $E_0$ based on a state observation: $\hat{p} = \lambda(s)$ for all state observations $s$ in $E_0$, and label prediction in the inner synthetic SL task based on a data instance: $\hat{y} = \lambda(x)$, for all $x \in X_d$. Once distillation is complete, a new learner $\lambda \in \Lambda$ trained on $\{X_d, Y_d\}_\theta$ should solve the environment $E_0$, or achieve an acceptable episodic reward on $E_0$ (without any direct training on $E_0$).

We define the synthetic data instances and labels to match the dimensionality of the states and actions of the RL environment, respectively. Thus, training the learner, the RL policy network, on the synthetic task is equivalent to learning to classify synthetic states into an action or regressing a state-dependent policy. However, the distiller is not guaranteed to generate data instances that are in or near the state space, nor label a reasonably state-like instance with a reasonable policy. The only constraint on the distiller is how well the dataset it produces can train the learners to perform on $E_0$.

The distiller only learns information pertinent to task learning. Thus, when we discuss distilling an environment, we mean that the learning trajectories and policies are distilled, not other aspects such as transition and reward functions.

### 3.1 PROXIMAL POLICY META-OPTIMIZATION

We consider PPO to be a great candidate for the distillation outer-learning objective due to the benefits of its trust-region protections. However, a direct adaptation of dataset distillation using PPO as the outer objective is not sufficient to utilize these trust region protections. PPO's trust region is created by the clip function, which sets the gradient to 0 when the change in policy throughout learning on a given RL trajectory exceeds a threshold set by $\epsilon$. This is because once clipping is applied, infinitesimal changes will not affect the result. This requires that the policy network $\pi_\phi$ be altered throughout training on the RL trajectory. However, only the synthetic dataset is changed by the policy loss during distillation. Thus, we alter dataset distillation such that the agents' initializations do not change throughout training on a given trajectory (i.e. over a single PPO epoch) to ensure that the only changes reflected in the change in policies is due to optimization on PPO loss, rather than factors such as parameter initialization or model architecture.

We propose a new formulation for task distillation using PPO as the outer objective, and introduce it as *Proximal Policy Meta-Optimization* (PPMO) for RL-to-SL distillation. This incorporates the policy trust region protections of PPO.

This process is detailed in Algorithm 1. Training is split into 3 nested loops: meta-epochs, policy epochs, and batched iterations, respectively. At each meta-epoch, a new agent/policy network initialization $\lambda_{\phi_{init}}$ is sampled and trained on the synthetic dataset (lines 2-3). This training uses mean-squared error loss on the model's prediction of synthetic data $X$ against the synthetic label $Y$, and we perform this learning in a single step of gradient descent. The model resulting from this inner-training, $\pi_\phi^0$, defines the policy that is run on the RL environment $E_0$, creating an RL trajectory $\tau$ (line 4). Just as in PPO, this trajectory is used for (outer) learning over multiple "policy epochs". For each batch drawn from the shuffled trajectory, PPO policy and value loss is calculated using

---

**Algorithm 1:** PPMO for RL-to-SL Distillation($\{X_d, Y_d\}_\theta$, $\Lambda$, $V_\psi$, $E_0$, $e$, $n$, $b$)

---

**input** : initialized synthetic dataset $\{X_d, Y_d\}_\theta$, learner distribution $\Lambda$, value network $V_\psi$, RL
  environment $E_0$, number of episodes per iteration $e$, number of policy epochs $n$, RL
  batch size $b$

**output:** learned synthetic dataset $\{X_d, Y_d\}_\theta$ distilled from $E_0$

**1 while** $\{X_d, Y_d\}_\theta$ *has not converged* **do**
**2**   $\lambda_{\phi_{init}} := \text{Sample}(\Lambda)$;
**3**   $\pi_\phi^0, \nabla_\phi := \text{Train}(\lambda_{\phi_{init}}, \{X_d, Y_d\}_\theta)$;
**4**   $\tau = \text{PerformEpisodes}(E_0, \pi_\phi^0, e)$;
**5**   $i := 0$;
**6**   **for** *policy epoch from* $1$ *to* $n$ **do**
**7**     **for** $(s, a, r) \in \tau$ *of batch size* $b$ **do**
**8**       $L_\pi, L_V := \text{PPOLoss}(\pi_\phi^i, \pi_\phi^0, V_\psi, s, a, r)$;
**9**       $\nabla\theta := \text{BackpropagateWithMetaGradients}(L_\pi, \theta, \nabla_\phi)$;
**10**       $\nabla\psi := \text{Backpropagate}(L_V, \psi)$;
**11**       Optimize $\theta$ and $\psi$ w.r.t. $\nabla\theta$ and $\nabla\psi$;
**12**       $\pi_\phi^{i+1}, \nabla_\phi := \text{Train}(\lambda_{\phi_{init}}, \{X_d, Y_d\}_\theta)$;
**13**       $i := i + 1$;
**14**     **end**
**15**   **end**
**16 end**
**17** return $\{X_d, Y_d\}_\theta$;

---

the formulas defined in Equations 1 and 2, taking in the current parameters of the agent $\pi_\phi^i$, the parameters of the agent used to gather the experiences $\pi_\phi^0$, the value network $V_\psi$, and the batch of experience data (line 8). Note that $A_t$ is calculated using GAE in our implementation. The policy loss is our meta-loss: it is backpropagated through the inner-learning process to update the synthetic dataset $\{X_d, Y_d\}_\theta$ (lines 9, 11). The critic is updated with standard optimization, as in standard PPO (lines 10-11). A new agent $\pi_\phi^{i+1}$ is trained from the same initialization $\lambda_{\phi_{init}}$ as the previous agents in this meta-epoch (line 12), and used in the next sampled batch. Once all trajectory batches have been iterated through $n$ times, the next meta-epoch iteration begins by sampling a new agent initialization (line 2). By the end of this process, the synthetic dataset can be used to train any agent initialization in $\Lambda$ to perform on $E_0$.

## 4 ND CART-POLE EXPERIMENTS

To control the difficulty of cart-pole and the size of its state and action spaces, we expand the cart-pole system into multiple dimensions. In ND cart-pole, the cart and pole have $N$ degrees of freedom, though the cart is limited to accelerating along one axis at each timestep, creating an action space of size $2N$. We utilize this extension of cart-pole to examine $k$-shot learning and minimum distillation size in RL-to-SL distillation. See Appendix B for a full description of ND cart-pole.

Our cart-pole experiments utilize a standard set of hyperparameters unless otherwise stated. We defined the learner distribution $\Lambda$ to contain a single architecture, $\alpha_c$, with random initialization. The architecture, initializations, and PPO hyperparameters used in this work are in line with the standard of Huang et al. (2022), except our removal of learning rate annealing and value loss clipping.

### 4.1 $k$-SHOT LEARNING

Once a synthetic dataset has been fully distilled, it can be used for $k$-shot learning. New agent models, including those not used in distilling, can be trained on the synthetic task using $k$ learning instances. In our experiments, these $k$ instances are learned as a single batch in one gradient descent step. As the distilled dataset was trained to teach agents sampled from $\Lambda$, these agents are expected to

| Experiment | $\Lambda$ | $\alpha_c$: Ortho std=1 | $\alpha_c$: Xe | $\alpha_c$: Xe std=1 | Random $h$ | Random $L$ |
|---|---|---|---|---|---|---|
| 1D 2-Shot | $500 \pm 0.0$ | $238 \pm 200$ | $258 \pm 207$ | $175 \pm 209$ | $500 \pm 0.0$ | $500 \pm 0.0$ |
| 1D 512-Shot | $500 \pm 0.0$ | $233 \pm 221$ | $246 \pm 198$ | $137 \pm 178$ | $500 \pm 3.8$ | $500 \pm 0.0$ |
| 2D 3-Shot | $365 \pm 31.9$ | $16.3 \pm 11.1$ | $76.8 \pm 67.3$ | $13.4 \pm 6.3$ | $443 \pm 55.7$ | $415 \pm 159$ |
| 2D 512-Shot | $350 \pm 37.2$ | $15.9 \pm 10.5$ | $68.6 \pm 50.9$ | $13.9 \pm 7.8$ | $457 \pm 81.0$ | $439 \pm 99.8$ |
| 3D 4-Shot | $264 \pm 20.6$ | $12.4 \pm 2.3$ | $38.6 \pm 21.5$ | $10.9 \pm 1.3$ | $379 \pm 93.3$ | $305 \pm 157$ |
| 3D 512-Shot | $240 \pm 26.1$ | $12.7 \pm 3.6$ | $41.8 \pm 24.0$ | $11.5 \pm 2.4$ | $401 \pm 132$ | $383 \pm 116$ |
| 4D 5-Shot | $127 \pm 7.4$ | $12.5 \pm 2.1$ | $35.0 \pm 12.2$ | $11.0 \pm 1.5$ | $165 \pm 29.8$ | $156 \pm 60.7$ |
| 4D 512-Shot | $129 \pm 6.8$ | $12.5 \pm 2.0$ | $39.7 \pm 13.2$ | $11.3 \pm 1.6$ | $203 \pm 55.7$ | $188 \pm 46.1$ |
| 5D 6-Shot | $87.5 \pm 4.3$ | $12.8 \pm 1.5$ | $35.7 \pm 9.2$ | $11.5 \pm 1.5$ | $112 \pm 18.3$ | $125 \pm 44.6$ |
| 5D 512-Shot | $87.5 \pm 4.0$ | $12.8 \pm 1.8$ | $41.0 \pm 7.1$ | $12.0 \pm 1.5$ | $140 \pm 40.1$ | $159 \pm 30.4$ |

Table 1: Mean and standard deviation of rewards over agents sampled from a variety of distributions, trained on distillation, then tested on ND cart-pole. The distributions are as follows: $\Lambda$ is the agent distribution used in training, consisting of a fixed architecture $\alpha_c$ and using orthogonal initialization with a standard deviation of $0.01$ on the final layer and $\sqrt{2}$ on the other layers. "$\alpha_c$: Ortho std=1" is the same as $\Lambda$ but with a standard deviation of 1 on all layers. "$\alpha_c$: Xe" uses architecture $\alpha_c$ but is initialized using Xe initialization with the same standard deviations as $\Lambda$. "$\alpha_c$: Xe std=1" uses Xe initialization with standard deviation = 1 for all layers. "Random $h$" modifies architecture $\alpha_c$ by randomly sampling the hidden layer size from the set $[32, \ldots, 256]$ (in $\alpha_c$ the size is fixed at 64). "Random $L$" modifies architecture $\alpha_c$ by randomly sampling the number of hidden layers (number of layers not counting the output layer) from the set $[1, \ldots, 6]$ (in $\alpha_c$ it is fixed at 2). Both Random $h$ and Random $L$ used the same initialization as $\Lambda$. Table 6 in Appendix D examines $h$ and $L$ values.

outperform other agents on the task after $k$-shot learning—outliers within $\Lambda$ and out-of-distribution agents may perform significantly worse on $T_0$ than the average performance of learners from $\Lambda$.

We verify the success of our distiller by performing $k$-shot learning on randomly sampled agent architectures and initializations in $\Lambda$ on the cart-pole environment (see summary of results in Table 1). Note that, because the agents in $\Lambda$ used in our experiments share the same architecture but are produced by randomly initializing the parameters, we can assume that no agent used in verification was used in training. We also explore deviations from $\Lambda$ to test $k$-shot learning on out-of-distribution learners. For each distribution tested, we sample 100 model architecture/initialization values, and train each model on the distilled dataset. The model is tested against the RL environment for 100 episodes, and the means and standard deviations of rewards over all the tested models is reported.

Our experiments showed little difference in $k$-shot learning based on the size of $k$: learning on the minimum-sized distillation (see Section 4.2) generally performed better than 512-shot with agents from $\Lambda$, but performed slightly worse than 512-shot with the other tested distributions. This implies that higher batch sizes are less likely to overfit to $\Lambda$. Our experiments show that $k$-shot learning on $T_d$ appears to be more sensitive to the parameter initialization function (see Ortho and Xe experiments) than to architecture changes (see Random $h$ and $L$ experiments). While this distillation could be used for neural architecture search due to its good generalization to architectures outside of $\Lambda$, it may still be biased to architectures close to those in $\Lambda$; using a wider array of architectures during training would ensure a fairer evaluation of varied neural architectures.

## 4.2 MINIMUM DISTILLATION SIZE

Distillation compresses a learning task, transforming it from many-shot learning to finite $k$-shot learning. $k$, the number of distilled instances to produce, is an important hyperparameter in distillation. Our experiments have determined that varying $k$ has only a small effect on the quality of the distillation, past a certain threshold. Increasing $k$ increases the computation costs of the distillation training non-negligibly, as all $k$ instances are updated in each outer learning step and a distiller with a larger $k$ requires more epochs on average to converge. However, the end-learning costs are negligibly increased, so long as $k$ fits into a single batch on the provided hardware.

There is, however, a minimum threshold $k_{min}$, such that distillation fails when $k < k_{min}$. This number is determined by the geometry of the label space (Sucholutsky & Schonlau, 2021a). The distilled instances and labels must distinguish the classes or actions from each other. This can be

done in fewer instances than the number of classes using soft labels. Using soft labels, the minimum number of instances to distinguish the classes of a label space with $c$ discrete values is:

$$k_{min} = \lceil c/2 \rceil + 1 \tag{3}$$

This allows for what Sucholutsky & Schonlau (2021a) call "less than one"-shot learning,[2] or learning on less than one instance per class.

Equation 3 was one of multiple demonstrated by Sucholutsky & Schonlau (2021a). We validate this equation experimentally on ND cart-pole and Atari. Distilling to $k \geq k_{min}$ instances results in a successful distillation—the expected reward for $\forall \lambda \in \Lambda$ trained on the distillation is approximately equal to the reward achieved by $\lambda$ after direct RL training. As an example, 2D cart-pole distillation achieves a mean reward of 350 when $k = 512$ and 365 mean reward when $k = k_{min} = 3$. When $k < k_{min}$, the distillation is unsuccessful in teaching learners to solve the task, resulting in low average rewards consistent with random performance in cart-pole. 2D cart-pole distilled to $k = k_{min} - 1 = 2$ achieves an average of 40.6 reward.

The equation $k_{min} = \lceil c/2 \rceil + 1$ determines how many instances are required to distinguish the classes of the action space, assuming all actions are necessary. In ND cart-pole, all actions are needed to solve the task, and all actions are performed approximately at the same frequency. Thus, the distilled dataset must provide distinction between each action in its soft labels. That is, there must be at least one instance in which the action probability label is high, and one where it is low, relative to the other action probabilities in the label.

See Figure 4 in Appendix C for a visualization of 1D and 2D cart-pole distilled to $k_{min}$ instances.

## 5 ATARI ENVIRONMENTS

We distill Atari environments to demonstrate RL-to-SL distillation scaling to more complex and difficult reinforcement learning tasks. The Atari environments have a much larger state and action space than cart-pole, as well as more complex reward functions, providing a significant challenge.

We have chosen a small selection of Atari tasks for brevity. We have avoided selecting tasks that are difficult for a standard PPO agent to learn, as finding tasks that can be learned by distillation but not by direct-task learning is beyond the scope of this work. As Atari tasks are mostly open-ended reward maximization problems, we determine the success of the distillation by comparing the average reward achieved by randomly sampled agent initializations from $\Lambda$ trained on a fully distilled dataset $\{X_d, Y_d\}_\theta$, against the average reward achieved by randomly sampled learners from $\Lambda$ trained directly on the environment using PPO. This is in line with the distillation ratio metric proposed by Sucholutsky & Schonlau (2021b), which also compares performance after direct-task learning versus distilled-task learning.

For the Atari experiments, we define $\Lambda$ as the distribution of random initializations of a single architecture $\alpha_a$, derived from the standardized PPO implementation of Huang et al. (2022), as with parameter initialization and all PPO hyperparameters. For the sake of feasibility, we limit the amount of training time for the baseline RL experiments. While it may be possible for the agents to achieve a higher reward given more training time, it is difficult to determine whether an RL agent has fully converged, or if it has reached a temporary plateau in performance; thus, we consider time limits and apparent convergence are a reasonable early-stop heuristic. In order to provide a more fair comparison to the converged full distillation results of *Centipede*, we trained the Centipede agent to 8000 epochs, giving it the same approximate number of update steps as in the fully distilled experiment. Those results are reported, yet note that the reward appeared to plateau at 1000 epochs. To provide a baseline for our RL agents themselves, we ensured our baseline PPO agents reached comparable performance to the results of DQN agents in the work of Mnih et al. (2015).

Due to the increase in complexity in Atari environments versus cart-pole environments, the cost of distillation increases greatly as well. It may be unreasonable to commit the large amount of required resources for a full distillation of an Atari environment without knowing how long distillation will take or if distillation can succeed given the experimental setup; thus, we provide a method for controlling the difficulty of distillation by providing intermediate steps between direct-task learning

---

[2]The term $k$-shot learning is not standardized. In this work, we define $k$ as the number of training instances.

---

**Algorithm 2:** Meta-gradient Task Distillation with Encoding($\{X_d, Y_d\}_\theta, \Lambda^l, \varepsilon_\xi^l, n, T_0$)

---

**input** : distilled dataset $\{X_d, Y_d\}_\theta$, learner distribution $\Lambda^l$, encoder $\varepsilon_\xi^l$, number of inner
        iterations $n$, target training task $T_0$ with data instances $X_0$ and loss function $L$
**output:** distilled dataset $\{X_d, Y_d\}_\theta$ and trained encoder $\varepsilon_\xi^l$

1 **while** $\{X_d, Y_d\}_\theta$ *and* $\varepsilon_\xi^l$ *have not converged* **do**
2    $\lambda_\phi^l := \text{Sample}(\Lambda^l)$;
3    $e_{X_d}^l := \varepsilon_\xi^l(X_d)$;
4    $\phi, \nabla_\phi, := \text{Train}(\lambda_\phi^l, e_{T_d}^l, Y_d)$;
5    $e_{X_0}^l := \varepsilon_\xi^l(X_0)$;
6    $L_{T_0} := L(\lambda_\phi^l, e_{X_0}^l)$;
7    $\nabla_\theta, \nabla_\xi := \text{Backpropagate}(\nabla_\phi, L_{T_0})$;
8    Optimize $\theta$ and $\xi$ w.r.t. $\nabla_\theta$ and $\nabla_\xi$;
9 **end**
10 return $\{X_d, Y_d\}_\theta$ $\varepsilon_\xi^l$;

---

and distillation. We do so by providing an algorithm that generalizes both direct-task learning and distillation. Then, we demonstrate this method on four Atari environments. We perform these experiments by distilling with $k = k_{min}$ for each of the four chosen Atari environments.

### 5.1 ENCODER ROLLBACK FOR VARIABLE DIFFICULTY

Due to the expense of distilling Atari games, we provide a method to scale the complexity of distillation training, providing evidence for the feasibility of full distillations. We utilize an encoder network to simplify the complex state spaces without reducing the dimensionality of the synthetic dataset. This formulation with the encoder as all but the final layer is defined and explored by Zhou et al. (2022) as a combination of dataset distillation and neural feature regression. While they use this formulation to train downstream task learners, we expand this method for difficulty scaling.

An encoder network $\varepsilon$ with parameters $\xi$ can be utilized to transform the learner model's input into a smaller space. We use our standard 5-layer Atari learner architecture $\alpha_a$ and split it along layer $l$, such that the encoder $\varepsilon_\xi^l$ consists of layers $[0, l)$ of $\alpha_a$ and the learner $\lambda_\phi^l$ consists of layers $[l, 5)$. The encoder's output is fed directly into the learner, just as if they were still a single network. The difference between learner and encoder lies in training: the learner's parameters are still updated by inner learning on the distiller's synthetic task $T_d$, and the learner is resampled from $\Lambda^l$ each outer iteration. The encoder, however, is only initialized at the start of training and is trained through outer meta-learning on $T_0$, rather than inner learning $T_d$. The encoder is trained alongside the distiller and the auxiliary critic network, rather than with the learner.

We provide the encoder formulation of generalized task distillation as Algorithm 2. The encoder is initialized with the distiller and, unlike the learner, is not reinitialized in the outer loop. The encoder produces encodings of the data associated with both the synthetic task (line 3) and the real task (line 5), which in RL-to-SL distillation includes the synthetic data instances and the environment's states, respectively. The learner predicts on the encoded data, rather than directly on the tasks' data (lines 4, 6). While the learner's parameters are updated through inner learning, the encoder's parameters are updated through outer learning, alongside the distiller (line 8). Note that the distiller could create synthetic data in the encoded space, rather than utilizing the encoder in inner learning. We utilize the encoder in inner learning, as that allows the distiller's output to be the same dimensionality throughout our encoder rollback experiments; thus, we can ensure that the distiller's output space is large enough to hold all information needed at any value of $l$.

While distilling with an encoder limits the use of the distillation to training only new learners that utilize that encoder, it does allow scaling the difficulty of distillation for tasks where a full distillation may be too computationally expensive. When $l = 0$, the encoder is 0 layers long, and thus training is equivalent to distillation without using an encoder, which we refer to as *full distillation*. When $l = 5$ (all layers of $\alpha_a$), the learner is 0 layers long, and thus the algorithm is equivalent to directly training

|  | RL | $l=4$ | $l=3$ | $l=2$ | $l=1$ | Full Distillation |
|---|---|---|---|---|---|---|
| Centipede | $8378 \pm 4238$ | $8438 \pm 758$ | $7860 \pm 756$ | $8266 \pm 561$ | $8144 \pm 614$ | $8042 \pm 492$ |
| Space Inv. | $1947 \pm 464$ | $2123 \pm 156$ | $1443^* \pm 245$ | $768^{*\dagger} \pm 63$ |  | $284^* \pm 2.1$ |
| Ms. P.M. | $3894 \pm 839$ | $3351 \pm 91$ | $2950^\dagger \pm 269$ | $2570^\dagger \pm 243$ |  | $415^* \pm 119$ |
| Pong | $17.3 \pm 3.8$ | $21.0^\dagger \pm 0.01$ | $5.6 \pm 6.4^{*\dagger}$ |  |  |  |

(a) Average End-Episode Reward Achieved at Convergence

|  | RL | $l=4$ | $l=3$ | $l=2$ | $l=1$ | Full Distillation |
|---|---|---|---|---|---|---|
| Centipede | 1000 | 3000 | 4000 | 5000 | 40,000 | 8000 |
| Space Inv. | 115,000 | 90,000 | 175,000* | 670,000*† |  | 205,000* |
| Ms. P.M. | 53,000 | 50,000 | 140,000† | 1,300,000† |  | 190,000* |
| Pong | 5000 | 17,000† | 50,000*† |  |  |  |

(b) Outer Epochs to Apparent Convergence

Table 2: Results of distillation training with $k = k_{min}$, rolling back an encoder from $l = 5$ to $l = 0$. † indicates that the run was performed with parallelization (6 gpus), so the number of epochs is not comparable to non-parallelized runs. ∗ indicates the distillation has not fully converged. Empty cells represent experiments that did not exceed random performance due to high computational costs.

the encoder network on $T_0$, i.e. direct-task learning. Since direct-task learning is generally more stable and computationally cheaper, higher $l$ values (larger encoders) represent easier distillation tasks, while lower $l$ values (smaller encoders) represent harder distillation tasks. Thus, if we can successfully distill a task with a given encoder, and show that we can roll the encoder back by reducing $l$ and successfully distill the task, we can show how the cost of distillation increases as $l$ decreases. In addition, we can provide evidence for the feasibility of performing full distillations for tasks for which the cost of distillation is prohibitive given fixed resources.

## 5.2 EXPERIMENTS

In the Atari experiments, a 5-layer agent (with architecture $\alpha_a$) is used. These agents are standard for PPO learning and are capable of learning the tested environments directly. We demonstrate that the environments can also be learned with an encoder with $l = 4$, where the learner consists of a single linear probe, and for decreasing values of $l$, at the cost of an increase in computational complexity. We test 100 agents on 100 episodes for each distillation experiment.

In general, as $l$ decreases, the number of outer epochs to convergence increases, though the amount by which it increases varies across the environments and network layers. This makes it difficult to accurately predict the cost of a full distillation; however, this method does provide a smoother increase in computation cost per step than jumping from reinforcement learning to distillation. We were able to fully distill *Centipede*, due to its relatively cheap cost of full distillation. The other environments quickly became prohibitively expensive for our resources; however, we assert that full distillations of these environments is possible given sufficient resources (see Table 2).

While these experiments provide evidence for the feasibility of full distillations of these environments, the costs may be prohibitive. However, distillation frontloads computation costs and allows for cheaper training on downstream tasks—as low as one optimization step per model. Given a large enough search space for neural architecture search, distillation is a cost-effective solution. Training Atari agents is expensive, and each agent must explore the environment on its own. Distilling an Atari environment allows for training these agents in a single optimization step without exploration.

For example, our distillation of the Atari *Centipede* environment took approximately 8 times the number of epochs to train than direct-RL on *Centipede*. The meta-optimization and inner training slightly increase the cost of each distillation epoch: on our hardware using one NVIDIA A100 GPU, distillation took on average 3.73 seconds per epoch versus 3.25 seconds per epoch for RL; distillation training took 1.15 times longer per epoch than direct RL training. The benefits of distillation come in training multiple models on the trained distiller. Training on the synthetic dataset is relatively cheap, taking one step of SGD on a single batch of data without any interaction with the environment. Each model takes an average of 0.18 seconds to train on the distilled task versus 3,250 seconds

| | Random | RL | Full Distillation |
|---|---|---|---|
| Ant | $-52 \pm 88$ | $4919 \pm 1488$ | $3115 \pm 319$ |
| Half Cheetah | $-282 \pm 79$ | $4486 \pm 1359$ | $2226 \pm 242$ |
| Hopper | $21 \pm 23$ | $2204 \pm 765$ | $2168 \pm 302$ |
| Humanoid | $123 \pm 37$ | $1602 \pm 1097$ | $544 \pm 32$ |
| Humanoid Standup | $34,034 \pm 3028$ | $212,265 \pm 86,359$ | $138478 \pm 6037$ |
| Inverted Double Pendulum | $57 \pm 16$ | $8781 \pm 1973$ | $1235 \pm 427$ |
| Reacher | $-42.95 \pm 4.05$ | $-5.23 \pm 1.83$ | $-7.78 \pm 0.57$ |
| Walker2D | $2.1 \pm 5.1$ | $4299 \pm 1156$ | $331.7 \pm 117$ |

Table 3: Results of 5 distillations on each of the MuJoCo continuous environments. PPO RL performance and uniform random agent performance are also recorded.

of RL, and requiring much less data: 10 instances versus 8,000,000 instances; 1 optimization step versus 15,625. While producing the distilled dataset takes 29,840 seconds with our resources, the entire process of training the distiller and performing $k$-shot learning on the distilled task is cheaper than direct-RL if one is training more than 9 models sequentially. To demonstrate how quickly this scales: in the time it takes to produce 10 *Centipede* agents, we can distill *Centipede* from scratch and initialize and train 14,777 agents on the distillation. For each additional model trained directly on *Centipede*, we could initialize and train an additional 18,055 agents on the distilled task in the same amount of time. If one is willing to pay the costs of distilling the dataset, they are rewarded with the ability to train 20,000 agents per hour without parallelization.

# 6 CONTINUOUS ENVIRONMENTS

Due to its importance in fields such as robotics, we distill continuous RL environments. We use the MuJoCo-v4 (Multi-Joint dynamics in Contact) environments (Todorov et al., 2012), which have continuous state and action spaces. Continuous PPO uses the state to predict the mean value of each action and learns a fixed log standard deviation for each action, independent of the state. The distilled dataset is treated similarly to the discrete environments. The distilled state trains the learner's mean action, while an action-agnostic standard deviation is learned. In addition, due to the continuous action space, we opted to distill to $k = 64$ instances, rather than determining $k_{min}$ through trial-and-error. We trained all models for 24 hours on 4 GPUs. Note that some models, such as the Inverted Double Pendulum and Walker2D may not have converged. We report our results in Table 3. The success of this method suggest that distillation can be used on a wide variety of RL tasks.

# 7 DISCUSSION

We have demonstrated distillation of cart-pole, Atari, and MuJoCo environments into compact, single-batch datasets. We have demonstrated its benefits in reducing training costs. While the up-front expenses of distillation are not insignificant, for applications that require training many models, the per-model speedup is significant. The cost may be reduced by using better reinforcement learning algorithms, better hyperparameters, and parallelization. The performance of distillation-trained models may be improved with a longer distillation process or a robust hyperparameter search. Performing more complex experiments on Atari and MuJoCo, such as $k$-shot learning on different learner distributions, could further increase our understanding of the principles explored in our cart-pole experiments; decreased distillation costs would increase the feasibility of such experiments.

Distillation has been proposed for use in downstream tasks such as data anonymity (Dong et al., 2022) and neural architecture search (Wang et al., 2018), and could be viable for other search problems, such as finding high-performance models or good parameter initializations. In addition, methods that require multiple end-task models, such as ensembling, may benefit from distillation. However, distillation has not yet been widely adopted for these use cases, likely due to applications being an afterthought in distillation research and to the computational expense of distillation. Generalized task distillation, and RL-to-SL distillation in particular, opens avenues for real-world application by expanding the types of problems that distillation can be used on. We intend for our future work to examine use cases of distillation to prove its viability for real-world applications.

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

## A APPENDIX - HYPERPARAMETERS AND ADDITIONAL DETAILS

| Distillation Hyperparameters | Values |
|---|---|
| distiller optimizer | Adam |
| critic optimizer | Adam |
| inner optimizer | SGD |
| distiller lr | $2.5 * 10^{-4}$ |
| critic lr | $2.5 * 10^{-4}$ |
| initial inner lr | $2 * 10^{-2}$ |
| outer objective | meta-PPO |
| inner objective | mean squared error |
| episodes per epoch | 10 |
| reward discount $\gamma$ | 0.99 |
| policy epochs | 4 |
| outer batch size | 512 |
| rollout length | 200 |

(a) Hyperparameters Used in Cart-pole and Atari

| | Cart-pole | Atari |
|---|---|---|
| state space | 4N | (4x84x84) |
| action space | 2N | 18 max (env dependent) |

(b) Hyperparameters that Differ in Cart-pole vs Atari

Table 4: Hyperparameters utilized in experiments.

| Environment | Number of Actions | $k_{min}$ |
|---|---|---|
| Centipede | 18 | 10 |
| Ms. Pac-Man | 9 | 6 |
| Pong | 6 | 4 |
| Space Invaders | 6 | 4 |

Table 5: Action spaces of Atari environments used.

## B ND CART-POLE

In an effort to create a reinforcement environment with a controllable state and action space, we have expanded the classic control cart-pole problem from a 1 degree-of-freedom problem to an arbitrary $N$ degree-of-freedom problem. This problem contains $N$ dimensions in which the cart can be moved and in which the pole can fall. The observation state consists of $N$ sets of quadruples: the cart's position, the cart's velocity, the pole's angle, and the pole's angular velocity in the corresponding dimension; or as 4 vectors of size $N$. We provide a diagram showing two views of 2D cart-pole in Figure 2

At each timestep, a force of magnitude $F$ is applied to the cart in a direction selected by the agent. To limit the action space to $2N$, we limit the applied force to $\pm F$ along one of the $N$ axes, resulting in a force of 0 along all other axes (other than gravity pulling the pole downward). This is similar to the classic 1D cart-pole problem, where the agent can select to apply a force of $\pm F$ to the cart along the dimension of the track. Another way of framing ND cart-pole is as $N$ independent cart-pole environments, in which the agent applies force $\pm F$ in exactly one environment per timestep, and the other environments receive a force of 0. If the agent fails in any of the parallel cart-poles, it fails in all. Like in standard cart-pole, each iteration provides a reward of +1 until the agent fails. In all ND cart-pole experiments, we terminate the environment once an accumulated reward of 500 is reached, an arbitrary reward ceiling often used in standard cart-pole (as unbounded cart-pole can lead to extremely long episodes).

This environment allows for easy difficulty scaling for reinforcement learning. However, the scaling does not appear to be linear. With a sufficiently high $N$, the agent cannot act in all dimensions

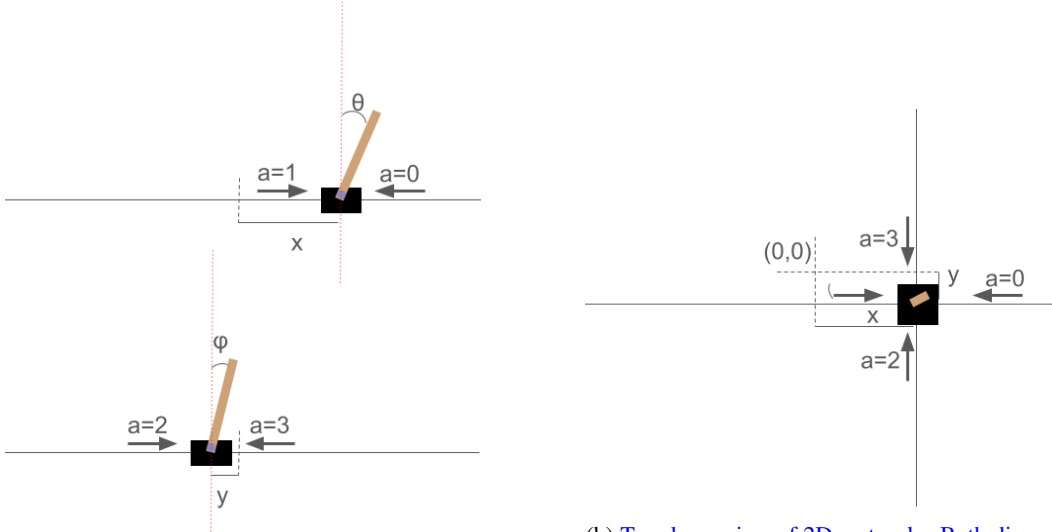

(a) Two perpendicular side views of 2D cart-pole. The solid lines represent the degrees of freedom of the cart.

(b) Top-down view of 2D cart-pole. Both dimensions representing the 2 degrees of freedom are visible. $\theta$ and $\phi$ are not clearly visible, as they represent the angle between the pole and the axes.

Figure 2: Two views of 2D cart-pole.

before the pole has a chance to fall in one of them. This can be counteracted by altering the time scale (i.e. scaling down velocity and acceleration), giving the agent a quicker reaction time. In our experiments, however, we only utilize up to 5D cart-pole, which can be solved by our PPO agent without changing the time scale.

# C ADDITIONAL FIGURES

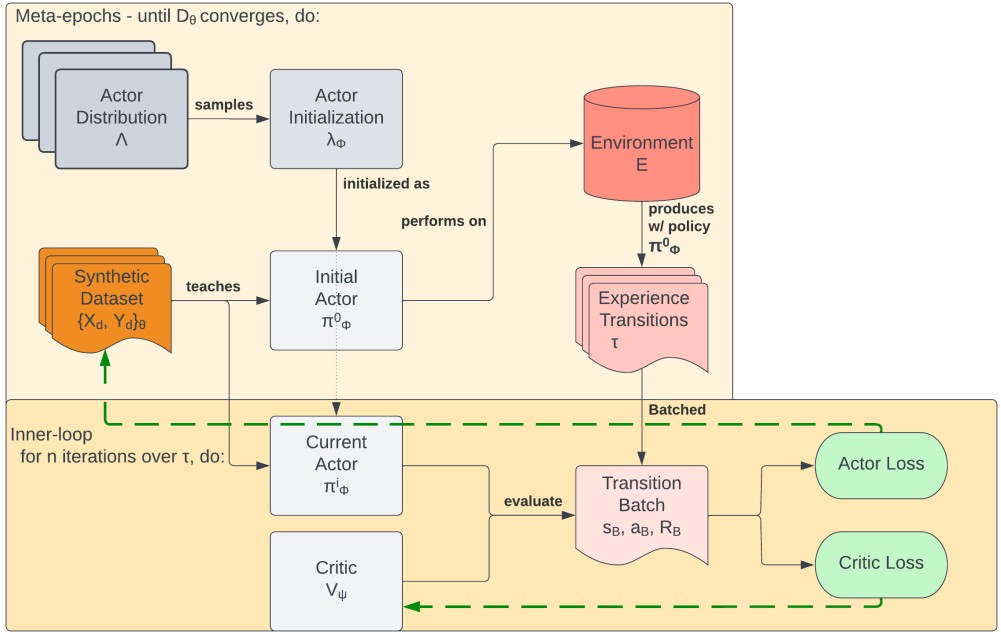

Figure 3: Training process for RL-to-SL distillation with generic actor-critic method.

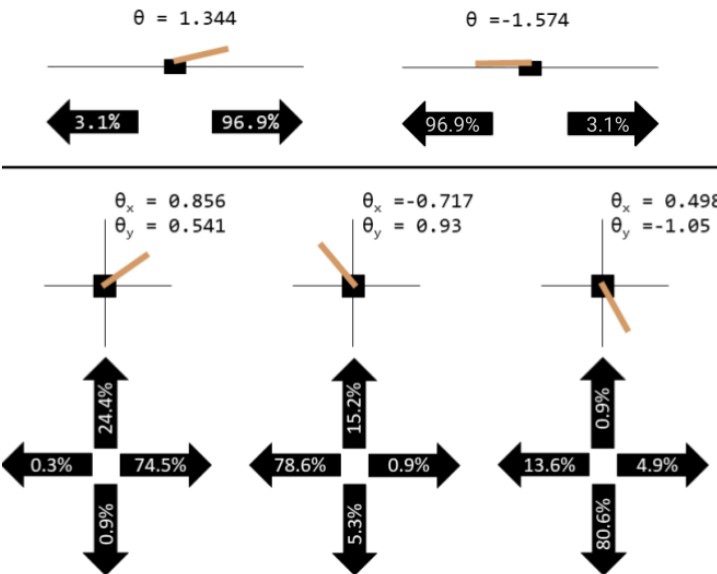

Figure 4: Visualizations of a distillation of 1D (above) and 2D (below) cart-pole with simplified information (only the angle and softmaxed labels are shown). While a full examination of interpretability is outside the scope of this work, this demonstrates that some tasks when distilled provide clear interpretability. In both distillations, the action label demonstrates that the agent should move the cart in the direction in which the pole is leaning. In 2D cart-pole, which was distilled to 3 states despite having 4 actions, the "up" action never has the highest probability mass in any of the labels; instead, the mass is split between two instances. This demonstrates how distillations with less than $c$ instances can teach a model to distinguish all $c$ classes or actions.

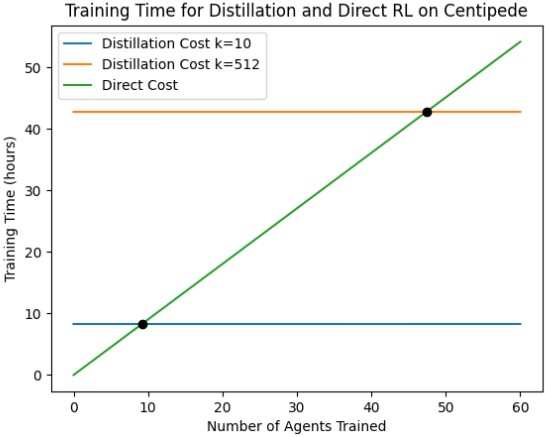

Figure 5: Comparing *Centipede* training time based on number of agents trained between direct-task learning (reinforcement learning), distillation with $k = 512$, and distillation with $k = 10$ ($k_{min} = 10$ for *Centipede*). Note that both distillation lines are parallel with a slope of $0.18$ seconds (the cost of training each learner with the final distilled set). Direct-task learning is cheaper than distillation training; however, training each agent after distillation using the distilled set is significantly cheaper than direct-task learning. Distillation training with $k = 10$ is significantly cheaper than distillation training with $k = 512$.

# D ADDITIONAL RESULTS

Here, we provide tables and figures with the results of additional experiments and results that did not fit into the main paper.

|       | h=32  | h=64  | h=128 | h=256 |
|-------|-------|-------|-------|-------|
| L=0   | 500.0 | 500.0 | 500.0 | 500.0 |
| L=1   | 500.0 | 500.0 | 500.0 | 500.0 |
| L=2   | 500.0 | 500.0 | 500.0 | 500.0 |
| L=3   | 500.0 | 500.0 | 500.0 | 500.0 |
| L=4   | 499.9 | 500.0 | 500.0 | 500.0 |
| L=5   | 494.9 | 499.5 | 500.0 | 500.0 |

(a) 2-shot 1D

|       | h=32  | h=64  | h=128 | h=256 |
|-------|-------|-------|-------|-------|
| L=0   | 94.4  | 90.4  | 93.1  | 93.2  |
| L=1   | 490.5 | 490.5 | 491.8 | 491.8 |
| L=2   | 500.0 | 500.0 | 500.0 | 500.0 |
| L=3   | 492.7 | 500.0 | 500.0 | 500.0 |
| L=4   | 447.8 | 490.4 | 499.2 | 500.0 |
| L=5   | 371.3 | 448.3 | 489.4 | 499.6 |

(b) 3-shot 2D

|       |       |       |       |       |
|-------|-------|-------|-------|-------|
| L=0   | 78.7  | 76.9  | 78.6  | 80.4  |
| L=1   | 430.7 | 468.7 | 482.7 | 488.2 |
| L=2   | 382.8 | 447.6 | 486.6 | 498.2 |
| L=3   | 292.7 | 368.4 | 441.5 | 488.1 |
| L=4   | 220.7 | 284.8 | 366.0 | 437.1 |
| L=5   | 170.7 | 221.8 | 287.0 | 374.9 |

(c) 4-shot 3D

|       |       |       |       |       |
|-------|-------|-------|-------|-------|
| L=0   | 49.1  | 50.7  | 51.1  | 50.9  |
| L=1   | 177.7 | 193.1 | 200.6 | 208.0 |
| L=2   | 182.0 | 210.2 | 235.7 | 261.0 |
| L=3   | 159.1 | 186.5 | 217.8 | 251.3 |
| L=4   | 135.6 | 162.1 | 189.6 | 225.2 |
| L=5   | 109.5 | 136.8 | 161.9 | 198.8 |

(d) 5-shot 4D

|       |       |       |       |       |
|-------|-------|-------|-------|-------|
| L=0   | 38.0  | 39.6  | 39.1  | 39.4  |
| L=1   | 125.0 | 134.5 | 140.0 | 143.4 |
| L=2   | 149.3 | 165.5 | 183.2 | 196.8 |
| L=3   | 135.1 | 153.8 | 175.1 | 197.5 |
| L=4   | 113.0 | 136.3 | 157.2 | 181.5 |
| L=5   | 79.5  | 115.1 | 136.0 | 160.9 |

(e) 6-shot 5D

Table 6: Extended results for Table 1, obtained by performing k-shot learning and validating on ND cart-pole, using a grid search over L ( $[0, 5]$ hidden layers) and h ($2^{[5,8]}$ hidden layer width). Error bars are omitted for brevity, but were of similar magnitude to those of Table 1.

|                | RL   | $l = 4$ | $l = 3$ | $l = 2$ | $l = 1$ | Full Distillation |
|----------------|------|---------|---------|---------|---------|-------------------|
| Centipede      | 8378 | 10,955  | 8251    | 7874    | 8327    | 7694              |
| Space Invaders | 1947 | 2941    | 1732    |         |         | 277*              |
| Ms. Pac-Man    | 3894 | 2987    | 731*    | 699*    | 342*    | 518*              |
| Pong           | 17.3 | 20.5    |         |         |         |                   |

(a) Average End-Episode Reward Achieved at Convergence

|                | RL      | $l = 4$ | $l = 3$  | $l = 2$ | $l = 1$ | Full Distillation |
|----------------|---------|---------|----------|---------|---------|-------------------|
| Centipede      | 1,000   | 1,500   | 3,000    | 2,000   | 55,000  | 40,000            |
| Space Invaders | 115,000 | 195,000 | 250,000  |         |         | 30,000*           |
| Ms. Pac-Man    | 53,000  | 110,000 | 55,000*  | 75,000* | 90,000* | 85,000*           |
| Pong           | 5,000   | 40,000  |          |         |         |                   |

(b) Outer Epochs to Apparent Convergence

Table 7: Results for $k = 512$ distillation, compare to Table 2.

| 1D | 2D | 3D | 4D | 5D |
|---|---|---|---|---|
| $21.8 \pm 11.6$ | $18.9 \pm 6.6$ | $18.5 \pm 4.9$ | $18.6 \pm 4.3$ | $18.6 \pm 3.8$ |

(a) ND cart-pole

| Centipede | Ms. PacMan | Space Invaders | Pong |
|---|---|---|---|
| $2443 \pm 1288$ | $239.6 \pm 80.3$ | $165.6 \pm 124.6$ | $-20.3 \pm 1.0$ |

(b) Atari

Table 8: Performance of uniform random agents.

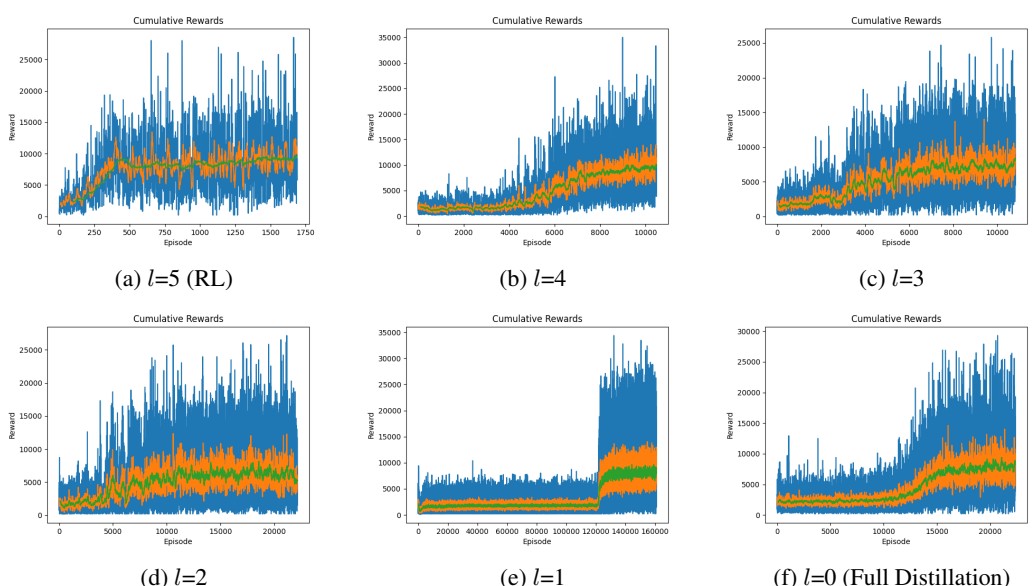

(a) $l$=5 (RL)  (b) $l$=4  (c) $l$=3

(d) $l$=2  (e) $l$=1  (f) $l$=0 (Full Distillation)

Figure 6: End-episode rewards for outer learning on *Centipede* distillation with $k = 10$ ($k_{min}$).

## E  DIFFUSION AS A DISTILLATION BASELINE

In order to provide a baseline for the distillation process, we trained a diffusion model to learn the state-action distribution of an expert Centipede RL agent, and used the data generated from the diffusion model to train RL agents.

We trained a single diffusion model until convergence on a dataset of 50,000 state-action pairs from the policy of the RL agent reported in our results.

We trained RL models on the diffusion-generated dataset. First, we trained an RL agent on 1600 generated pairs (taking approximately 19 hours, approximately doubling the time requirements of distillation training). This agent was tested over 1000 episodes on Centipede and got an average reward of 2067.

We then tested a cheaper method of generating a single-batch dataset of 32 instances (2 per action class) and allowing the models to train until convergence. This more than triples the data usage of distillation. We tested this on 5 agents over 1000 episodes each and got an overall mean reward of 1708; the best performing model got a mean reward of 1980.

The Centipede distillation, with a mean reward of 8083 far surpassed these baselines. It beat the first in compression size and time, including distillation training. The second method was much faster to train than to train a distillation, but the end-training of distillation is significantly faster and uses less data. These results demonstrate that distillation training is not easily replaceable with generative AI methods trained on a near-optimal state-action distribution.

