# OpenReview forum: "Distilling Reinforcement Learning into Single-Batch Datasets"
_ICLR.cc/2025/Conference — Submitted to ICLR 2025_

### Official Review · Reviewer_P1g3 · 2024-10-31

**Soundness:** 4
**Presentation:** 4
**Contribution:** 3
**Rating:** 8
**Confidence:** 5

**Summary:**

This paper proposes a new method / framework generalizing dataset distillation approaches to the reinforcement learning setting. A network distills the "optimal juice" from an RL environment. During test time, this distilled network can generate a dataset (input-output pairs) such that a supervised learning model trained on this dataset will get optimal rewards. Intuitively, I can think of the distilled network as trying to memorize the state-action stationary distribution of the optimal policy of an environment.

**Strengths:**

1) I think this paper is tackling a novel problem in a novel way. Its "freshness" is definitely a strength.
2) While the results obtained right now are not ground breaking, it does open up avenues for future work.

**Weaknesses:**

1) Missing baselines -- As far as I can tell, there are no baselines used in the paper. One fairly obvious (but could be strong) baselines is to train a PPO agent till convergence in the environment and use a diffusion model to learn the state-action distribution of the expert policy. At meta-train time, a newly generated agent can just use the state-actions generated by this model as a training dataset.

2) Motivation -- The motivation behind why this line of work is important is not clear to me after reading the introduction. Let me elaborate -
    a) Hyper-parameter / architecture tuning - Do you believe that the hyper-parameters tuned using the distilled network will be robust when training on a new environment without distillation. It do not know, nor is there any evidence provided in the paper, that hyper-parameters / architectures that work well using distilled network will work well when training in a new RL environment from scratch.

3) Data anonymity -- I think readers might appreciate of when this is a compelling reason in the RL setting.

**Questions:**

See Weaknesses section for points / questions to address.

---

> ### Author Response · Authors · 2024-11-15
>
> Thank you for your review. Let us know if the following answers your questions and concerns.
>
> W1: We compare the success of distillation-trained agents against the performance of comparable RL agents. Rather than representing a baseline (a floor for good performance), this represents a ceiling for good performance. That is to say, we would not expect the distillation to regularly outperform RL models, as these models are used in the distillation process, so the performance of the regular RL models represents an upper bound on what we can expect from the distillation-trained models. Because the models trained on distillations are shown to achieve nearly the same performance as these RL models, we consider this significantly stronger evidence of success than showing improvement over a baseline.
>
> W2: We consider the ability to significantly compress a large, complex task into a compact representation several orders of magnitude smaller than the original to be intrinsically useful. We distill Centipede, which required learning on 8,000,000 instances for our standard PPO models, into just 10 instances. Rather than taking nearly an hour to train a PPO model, training on the distillation takes 0.18 seconds. We consider this dramatic reduction of cost to be notably valuable in an age where deep learning costs continue to increase.
>
> While we agree that many hyperparameters will not be pertinent between distillation-training and RL-training, we have demonstrated evidence that the distillation is capable of generalizing to architectures not used in training (see Table 1). While we do not demonstrate a neural architecture search here, we assert that these findings strengthen the argument that neural architecture search is a valid use-case for distillation. Even if the distillation overfit the architectures used in training, we could instead produce a distillation using the neural architecture search space to ensure a relatively fair evaluation of these architectures.
>
> We point you towards other distillation works that perform neural architecture search experiments: Such et al. (2020) [1] demonstrate that there is a high correlation between high-performing architectures of distillation-trained and standard-trained models. In their three works, Zhao, Bilen, [2,3] (and Mopuri in [4]) demonstrate neural architecture search using their proposed distillation methods.
>
> W3: We used data anonymity as a use-case for dataset distillation in general. While we admit that anonymity may not apply as clearly for RL as it does in SL, there may be some use cases, such as when an environment’s mechanics are proprietary or expensive to simulate. Offline RL also does not simulate the environment and does not require details on the environment’s mechanics, yet is a popular RL strategy. We consider the use-case of distillation in regards to anonymity to be the same as offline RL, with the addition of distillation not requiring the explicit storage of specific trajectories.
>
>
>
> [1] Felipe Petroski Such, Aditya Rawal, Joel Lehman, Kenneth O. Stanley, and Jeff Clune. Generative teaching networks: Accelerating neural architecture search by learning to generate synthetic training data. In Proceedings of the 37th International Conference on Machine Learning, pp. 9206–9216, 2020.
>
> [2] Bo Zhao and Hakan Bilen. Dataset condensation with differentiable siamese augmentation. In International Conference on Machine Learning, 2021.
>
> [3] Bo Zhao and Hakan Bilen. Dataset condensation with distribution matching. In 2023 IEEE/CVF Winter Conference on Applications of Computer Vision, pp. 6503–6512, 2023.
>
> [4] Bo Zhao, Konda Reddy Mopuri, and Hakan Bilen. Dataset condensation with gradient matching. In Proceedings of the 9th International Conference on Learning Representations, 2021.

---

> > ### Comment · Reviewer_P1g3 · 2024-11-19
> >
> > > We consider the ability to significantly compress a large, complex task into a compact representation several orders of magnitude smaller than the original to be intrinsically useful
> >
> > I am unsure if this what the algorithm is doing exactly. I do not think that the distilled network has compressed any other aspect of the environment, other than the trajectory distribution of a near optimal policy. I think it is possible that the distilled network is just learning the state-action distribution of the optimal policy. At meta train time, it generate samples from this distribution and the new network just learns the mapping akin to behavior cloning (model based version of behavior cloning).
> >
> > The baseline I was talking about is not a standard RL agent, but a model of the state-action distribution of the optimal RL agent. The untrained meta-train time network will be trained by distillation from data generated by this model.

---

> > > ### Author Response · Authors · 2024-11-20
> > >
> > > Thank you for your continued attention and your response.
> > >
> > > **Environment Compression**
> > >
> > > We think there may be a misunderstanding with what we are claiming. We are not claiming that the entirety of the environment is being compressed. It is clear from our approach that aspects of the environment such as state transitions and reward functions are not explicitly stored in the distillation. We do not intend to claim that the distillation represents the environment completely; rather, that it represents a "core" of the training task that allows for models to learn the task in a single step, rather than the entire RL process. That is to say, when we refer to distilling an environment, we intend that we are distilling the reinforcement learning process on the environment. We recognize that certain optima of the task may not be learnable by the distillation, but we demonstrate that on average the optima that are learnable by the distillation approach RL performance. If you believe we misrepresented this position in the paper, we can do another revision to clarify this position.
> > >
> > > If we assume that our approach only learns the state-action distribution of a near-optimal policy, we have still compressed an 8 million instance RL problem into a 10 instance one-step SL problem. That is, our method provides a replacement for RL, which also finds only a single near-optimal policy. While having a method to prove that the distillation is capable of teaching various different state-action distributions would be valuable, we argue that even compressing a single near-optimal policy is inherently valuable, in addition to the use cases of distillation stated in the paper.
> > >
> > > If you know of a good method for proving that the policies between two given agents are reasonably distinct, with a good measure of what reasonably distinct means, we are willing to implement it to better verify our distillation. We do not believe measures such as distance in parameter space or differences in RL trajectories clearly demonstrate this.
> > >
> > > **Baselines**
> > >
> > > Methods such as model/policy distillation, behavior cloning, and the method you suggested can replace the original training process, but require a much longer learning process than training on the distilled dataset. The salient benefit of our solution is the single-step learning, which no other existing method we know of can provide. Altering any other method to do so would be non-trivial. The baselines proposed can at most be expected to match RL performance and to take longer than training on the distilled dataset; thus, we consider the RL performance "ceiling" to be an appropriate measure of our method's success.
> > >
> > > That being said, we are implementing the baseline you suggested and have trained a diffusion model on the state-action distribution of an expert Centipede agent. The high time costs of the diffusion generation process will limit the amount of synthetic data we can train on. We apologize that a full and fair examination of this baseline may not be possible during the review period given these time costs, yet we will do what we can. We will report our results in another response, and if they are insightful we will add them to the paper.

---

> > > > ### Comment · Reviewer_P1g3 · 2024-11-22
> > > >
> > > > Thank you for responding to my comments!
> > > >
> > > > > We are not claiming that the entirety of the environment is being compressed. It is clear from our approach that aspects of the environment such as state transitions and reward functions are not explicitly stored in the distillation
> > > >
> > > > I do think this is a point which does not come across to me clearly during the first read. The paper does make the claim that it compresses / distills the environment, which is somewhat vague and can be easily misunderstood to mean it compresses the transition dynamics / rewards or other important aspects of the environment.
> > > >
> > > > > If we assume that our approach only learns the state-action distribution of a near-optimal policy, we have still compressed an 8 million instance RL problem into a 10 instance one-step SL problem
> > > >
> > > > I am not dismissing the importance of this, I just think there are much simpler / obvious baselines (the one I described in the previous review) which I think would work well. And I think this baselines should be included in the paper.
> > > >
> > > > > If you know of a good method for proving that the policies between two given agents are reasonably distinct, with a good measure of what reasonably distinct means, we are willing to implement it to better verify our distillation
> > > >
> > > > A very simple way of measuring the difference is to calculate the KL divergence between the policy1 and policy2 empirically.
> > > > $$E_{p^{\pi_1}( s ) } [ \frac{ \log \pi_1(a \mid s) }{ \log \pi_2(a \mid s)  } ]$$
> > > >
> > > > > The salient benefit of our solution is the single-step learning, which no other existing method we know of can provide
> > > >
> > > > I agree that your approach will train considerably faster than the policy distillation baseline which I mentioned above. But if speed is a major contribution, it should be explicitly stated with comparison against baselines.
> > > >
> > > > > we consider the RL performance "ceiling" to be an appropriate measure of our method's success
> > > >
> > > > I agree that the RL performance is a "ceiling". But it can still be beneficial for the readers to know if other simple baselines can also reach this "ceiling" performance on the tasks you test your algorithms on. This would suggest for example, that more complex tasks are required to differentiate the performance of your method. It could also be the case that with more complex environments, the simple distillation baseline outperforms the algorithm proposed in your paper. This is just speculation of-couse, but without concrete comparisons there is nothing more the readers can do.

---

> > > > > ### Author Response · Authors · 2024-11-24
> > > > >
> > > > > Thank you for continuing to communicate with us to help us clear up these issues and strengthen our paper. We will submit a revision tomorrow with these additions. Please note that our resources and time are limited for this review period, so we will have to prioritize some experiments over others, but we will continue to run experiments up to the camera-ready version.
> > > > >
> > > > > **Distilling the environment vs policies:**
> > > > >
> > > > > Your point is fair. In our next revision, we will clarify the language to ensure our claims about what we are distilling to not exceed what we are actually distilling.
> > > > >
> > > > > **Policy differences:**
> > > > >
> > > > > This method should work well. We will have to ensure that we gather states and actions from both policies to ensure a fair evaluation. We can begin to implement this to compare the distances between different distillation-learned policies, diffusion-baseline policies, and different RL-learned policies to determine if a single distillation can produce multiple polices.
> > > > >
> > > > > **Diffusion baseline:**
> > > > >
> > > > > We have run some experiments to implement this baseline with Centipede. We are limited in resources and time, especially this late in the reviewing period, but we have attempted to give it a fair shot. We will continue to implement baselines for the other environments as well.
> > > > >
> > > > > We trained a single diffusion model until convergence on a dataset of 50,000 state-action pairs from the policy of the RL agent reported in our results. Then, we trained RL models on the diffusion-generated dataset.
> > > > >
> > > > > First, we trained an RL agent on 1600 generated pairs (taking approximately 19 hours, approximately doubling the time requirements of distillation training). This agent was tested over 1000 episodes on Centipede and got an average reward of 2067.
> > > > >
> > > > > We then tested a cheaper method of generating a single-batch dataset of 32 instances (2 per action class) and allowing the models to train until convergence. This more than triples the data usage of distillation. We tested this on 5 agents over 1000 episodes each and got an overall mean reward of 1708; the best performing model got a mean reward of 1980.
> > > > >
> > > > > While our Centipede distillation clearly beats this baseline with a mean reward of 8083, it is also possible that the baseline could be improved with better diffusion models or more training time. However, we have been more generous with time and data allotments to these diffusion datasets than what our Centipede distiller required. We will add these preliminary baseline results to the paper, and will continue to explore the second (less expensive) method to provide baselines in the other experiments.
> > > > >
> > > > > We will also clarify the contribution of speed and data compression more clearly in our results.

---

> > > > > > ### Author Response · Authors · 2024-11-30
> > > > > > **Baseline results**
> > > > > >
> > > > > > We implemented your proposed baseline for the other environments as well. Here are the results:
> > > > > >
> > > > > > - 1D Cart-Pole: $9.3514 \pm 0.0224$
> > > > > > - 2D Cart-Pole: $11.7564 \pm 0.8818$
> > > > > > - 3D Cart-Pole: $14.7310 \pm 2.7599$
> > > > > > - 4D Cart-Pole: $9.5976 \pm 0.4184$
> > > > > > - 5D Cart-Pole: $14.4210 \pm 3.7190$
> > > > > > - Ms. PacMan: $116.0360 \pm 102.1699$
> > > > > > - Pong: $-20.9946 \pm 0.0108$
> > > > > > - Space Invaders: $127.4460 \pm 102.0371$
> > > > > >
> > > > > > We hope that this satisfies your issue with a lack of baseline. Please let us know if we have satisfied all your questions and criticisms, or if there is anything else we can do.

---

> ### Comment · Reviewer_P1g3 · 2024-12-02
>
> Thanks for including these results. It is surprising to see that these results are so low, I would have expected them to perform better. Regardless, I have increased my confidence, soundness and the contribution score.
>
> It would be awesome if the authors include these results in the main paper and also open source the code for them!

---

> > ### Author Response · Authors · 2024-12-02
> >
> > We will definitely include these results in some way in the final version of the paper.  Thanks very much for all your engagement, feedback and suggestions---we enjoyed the interactions and the paper is better for it.  Thanks for recognizing that and increasing those scores to reflect it.  Finally, based on those increases, would you consider also increasing your overall rating to perhaps better align with the new scores?

---

### Official Review · Reviewer_AwMk · 2024-11-02

**Soundness:** 3
**Presentation:** 2
**Contribution:** 3
**Rating:** 6
**Confidence:** 4

**Summary:**

This paper introduces a novel distillation framework for distilling online RL environments to a synthetic dataset, which allows performant agents to be trained with one-step gradient descent. It extends PPO to leverage it to produce synthetic examples. Experiments are conducted on the cart pole environment and its n-dimensional extensions. Moreover, a few atari experiments are also performed to demonstrate its generalizability.

**Strengths:**

* Work is novel considering it is probably the first to introduce dataset distillation for online RL. However, there is a related work on dataset distillation for offline RL that is missing in related work and I believe should be discussed.(https://arxiv.org/abs/2407.20299)

* Results are promising in both Cartpole and atari experiments

* Framework does not introduce a significant overhead to the wall-time of the PPO.

**Weaknesses:**

* One of the main issues of this paper is motivation. Dataset distillation is proposed so that a large dataset can be condensed into a synthetic smaller one however I dont think the same analogy is reasonable for RL. Firstly, when you train a supervised model with large datasets, you would get almost the same accuracy which means that this dataset would have a score equivalence given a model whereas this is not accurate for the RL environments because RL environments are not deterministic. So, RL agents' performance varies across different numbers of runs (Agarwal et al., 2021). All in all, the distilling of a single instance of an environment will only represent the data that has been generated by this specific environment, not the environment itself. I would be very surprised if the policy generated by synthetic examples will generalize to the other instances with different seeds. Lastly, the discussion of runtime and the training of 10-20k agents is problematic due to similar reasons. You should get a unique agent if all are initialized the same however RL agent would be different if the environment is not fixed. I believe the randomness of the agents is due to the randomness of initialization.



* Evaluation setup is vague, in particular in Atari. In table 2(a), you present $\textit{Average End-Episode Reward Achieved at Convergence}$. Is it the average total reward of the last 100 episodes, and also what does the st dev mean for this table, is it the average over multiple runs? I believe the evaluation setup should be clarified to interpret the given results.


* More runs and different environments(could be more atari games, or continuous control env like mujoco) are needed to demonstrate the effectiveness of the proposed method. No need for full distillation for new experiments, l4 is good enough.


* Paper is a bit hard to parse(especially section 3.1), so writing could be improved. I believe Figure 2 is redundant, does not provide any insights, and is not visually appealing. Algorithm 1 is sufficient to get the concept. Lastly, you could import booktabs package for a more used format by the ICLR papers.





References
- Agarwal, Rishabh et al. “Deep Reinforcement Learning at the Edge of the Statistical Precipice.” Neural Information Processing Systems (2021).

**Questions:**

Q1) How does the policy induced by the synthetic examples preserve the trust region since the policy is not updated by the PPO loss?

Q2) Could you provide visualizations of the learned synthetic examples for 2D cartpole examples?

Q3) Do you have any insights regarding why lower ks are better even in higher dimensional cartpole settings?

---

> ### Author Response · Authors · 2024-11-15
>
> Thank you for your review. Here, we will reply to the specific weaknesses and questions in your review, and note where we will add revisions to the paper. The first revision will take into account all the feedback from the reviewers, with later revisions as further requested experiment results come in.
>
> Let us know if this answers your questions and concerns, or if there are ways we can better address them.
>
> W1: answered in separate comment due to space constraints.
>
> W2: It is the average reward of 1000 agents trained on distillation, each performing one episode. This evaluation is performed after training, rather than using the rewards gathered during training. The mean and standard deviation of these runs are reported. We will be more explicit in explaining our evaluation in our first revision of the paper.
>
> W3: We have begun experiments with the MuJoCo environments and will publish the results before the end of the discussion period.
>
> W4: We will revise section 3.1 for clarity. Do you have any specific concerns about that section or others?
>
> We will move Figure 2 to the appendix; we’ve had conflicting opinions about whether it is useful or not.
>
> We will look into booktabs to see if the new format improves table readability.
>
> Q1: This is described in the second paragraph of 3.1. In short, the trust region depends on the fact that PPO actor loss is the only factor changing the policy each iteration. Since distillation meta-learning uses PPO to update the distilled dataset rather than the policy, we must ensure that the resulting policy changes only due to the update to the distilled dataset. Thus, we reinitialize the agent to the same agent initialization used throughout an epoch of PPO. This differs from standard distillation, which reinitializes the model randomly after every update to the distilled dataset.
>
> Q2: Yes, we will provide these visualizations in the appendix in the first revision.
>
> Q3: The largest factor is likely that lower k values means there are less parameters to update in distillation meta-learning, making learning much quicker (compare Table 6 in the appendix to Table 2). In all our experiments, we’ve found that changes in k (above $k\_{min}$) do not affect performance much, but do affect training costs significantly. The differences reported between $k=k\_{min}$ and $k=512$ are not significant, and in examining other distributions in $k$-shot learning, $k=512$ sometimes does better and sometimes does worse.

---

> > ### Comment · Reviewer_AwMk · 2024-11-24
> >
> > W2) The standard procedure is to have at least 100 episodes for a single evaluation. Do you have these results? Are 1,000 agents evaluated on 1,000 distinct ends?
> > W3) Mujoco results strengthen the paper but it would be better to add random scores as well to see the improvement with distillation.
> > W4) It is hard to understand verbally so it would be better if you reference how they relate to the algorithmic step in Algorithm 1.
> >
> > Additional question:
> > Furthermore, how many runs are performed for PPO?

---

> > > ### Author Response · Authors · 2024-11-24
> > >
> > > Thank you for your continued responses. We believe we misunderstood some of what you stated in your review. We will attempt to clarify that here. Please let us know if we misinterpreted your review again.
> > >
> > > As it is late in the authors/reviewers discussion period and our resources and time are limited, we cannot perform all the requested experiments by the end of the discussion period. We will prioritize the most important experiments during this period to strengthen our paper as much as possible.
> > >
> > > **Environmental Randomness**
> > >
> > > We utilize the OpenAI gymnasium environments. After each episode of gathering data on the environment, the environment is reset. After each reset, the environment is randomized, so as training of a single model occurs, the environment is continuously randomized. If we consider each randomization of the same environment class a separate environment, training occurs over multiple environments, never using the same one for multiple episodes.
> > >
> > > If this is what you are referring to when you say different environment instances, then we considered it to be standard practice in RL, so we not did believe it needed addressing. It is the default behavior for the OpenAI gymasium environments we utilized. We can revise the paper to address this. If this is not what you are referring to, we are still unclear as to what it means to vary the randomness, as the sources of randomness we see in the environment are varied between resets.
> > >
> > > You are correct that we have only performed one distillation and evaluated the single distillation used to train 1,000 agents. The only source of randomness between these agents in training is the initialization, but during evaluation, each agent is given a separately-randomized environment. We agree that repeat distillations would strengthen our results.
> > >
> > > It is likely that the distillation, which is trained by PPO meta-learning, would be influenced by the same influences as PPO. However, we do not agree that this implies that this invalidates the use-cases of distillation. Dataset distillation (distillations of supervised datasets) has found its own niche apart from standard SL due to the compression and reduction of training time provided by the distillation. We consider the compression and training time reduction to be inherently valuable contributions, in addition to the use-cases mentioned in the paper. Distillation provides separate benefits to standard deep learning; they do not occupy the same niche.
> > >
> > > W2: Each of the 1,000 agents is evaluated on a single episode. While this does not provide a fair comparison of the individual agents, we believe this provides a fair evaluation of the distillation. We considered this sufficient to overcome the environmental randomness at evaluation time, as the distillation is compared over 1,000 episodes. Were we to make any claims about an individual agent's performance, we agree that it would require evaluation over multiple episodes.
> > >
> > > W3: Do you mean to provide the rewards of performing random actions on each environments to establish a baseline? We can perform those experiments and add those to the paper.
> > >
> > > W4: We've had conflicting feedback about whether the text should provide a step-by-step overview of the algorithm. We will attempt to find a middle ground, using the text we currently have with additional line number references where appropriate.
> > >
> > > Additional: As stated in the paper, we used only one RL agent. We did check the results with the DQN performance provided by Mnih et al. (2015) in "Human-level control through deep reinforcement learning." to ensure the results were reasonable. We agree that repeated RL training is vital for evaluation and we will attempt to get those repeated trials performed and recorded by the end of the reviewing period, at least for the Centipede environment.

---

> > > > ### Author Response · Authors · 2024-11-24
> > > > **Hopper distillation repeat trial results**
> > > >
> > > > We do actually have repeat trials for the Hopper distillation as we were testing saving/loading/validation on our system. The five agents received the following rewards and std devs over 100 trials:
> > > >
> > > > $2566 \pm 1085$; $1924 \pm 1054$; $1802 \pm 1011$; $2478 \pm 812$; $2071 \pm 1089$
> > > >
> > > > This is an overall mean of $2168$ and the standard deviation of the 5 means is $302$.
> > > >
> > > > We will continue to do repeat trials for the other MuJoCo distillations, as that is the most feasible given our resources and time limit. We will add the results in subsequent revisions before the end of the discussion period.

---

> > > > ### Comment · Reviewer_AwMk · 2024-11-24
> > > >
> > > > By "environment randomness," I am referring to the initial RNG seed provided to the environment. A reset does not constitute a new environment, as it uses the same RNG seed for sampling. The discussion regarding distillation pertains to its performance when PPO fails to learn an effective policy. If distillation continues to produce favorable outcomes in such scenarios, it would be advantageous. However, as you suggested, this may not be the case.
> > > >
> > > > I am unclear why evaluations are not conducted over 100 episodes for each agent. Evaluation is not computationally expensive, and such an approach is considered standard practice, as outlined in the original PPO paper. For more robust evaluation, I recommend sampling approximately 50 distinct agents generated through SGD and evaluating them across at least 50 different environments. Each agent should be assessed over at least 100 episodes per environment to account for variability in the results. The evaluation methodology employed in your work raises concerns based on these points.
> > > >
> > > > W3 -> Yes
> > > >
> > > > Overall, I consider the work to be valuable; however, I find the evaluation setup to be questionable, and the absence of multiple runs presents a significant limitation to the paper. Nonetheless, I have increased my score to 5, and it is likely to remain at this level unless the concerns are addressed.

---

> > > > > ### Author Response · Authors · 2024-11-24
> > > > >
> > > > > Thank you for your responses and for increasing the score. We appreciate your help in strengthening our paper, and we apologize for our misunderstandings of what you've been describing.
> > > > >
> > > > > **Distillation Evaluation**
> > > > >
> > > > > We agree with your evaluation method; it is both inexpensive and provides a stronger evaluation. We will prioritize using this method to evaluate our currently-held results. We will ensure this evaluation procedure results are in before the end of the discussion period. We will continue to attempt to implement 5 distinct runs for RL and distillation evaluations as well, but we will focus our resources on your proposed evaluation method for the moment.
> > > > >
> > > > > **W3**
> > > > >
> > > > > We have completed those experiments with MuJoCo and added them to the table. We plan to upload our next revision containing these results today. We will also add random agent performances of the ND cart-pole and Atari environments in the appendix (as those tables don't have space).
> > > > >
> > > > >
> > > > > **Environment RNG**
> > > > >
> > > > > As far as resetting not constituting a new environment, we examined this and we disagree that resetting reinitialized the RNG with the same initial seed as the prior run. According to the gymnasium documentation for `env.reset`:
> > > > >
> > > > > "This method generates a new starting state often with some randomness to ensure that the agent explores the state space and learns a generalised policy about the environment. This randomness can be controlled with the seed parameter otherwise if the environment already has a random number generator and reset() is called with seed=None, the RNG is not reset."
> > > > >
> > > > > In other words, the RNG's seed is not reset to its initial seed. Rather, the seed continues to advance from the last point it was in. Unless we assume that NumPy's random number generator reaches a fail state, the random numbers sampled will not be the same between resets, and thus each epoch in both training and evaluation uses a unique sequence of random numbers. Thus, explicitly using a new seed with not improve randomness between resets, unless the NumPy RNG is in a failure state.
> > > > >
> > > > > Testing this with a single CartPole-v1 environment, we get an initial state of `[-0.03816158, -0.04486908,  0.00071113, -0.0391084 ]` and after one reset an initial state of `[ 0.0380632 , -0.00080108,  0.04480701,  0.02464667]`. Thus, unless the RNG is in a failure state, each reset constitutes a unique environment instance and is equivalently random as randomly sampling a new random seed each reset.

---

> > > > > > ### Author Response · Authors · 2024-11-25
> > > > > > **First results from new validation**
> > > > > >
> > > > > > We ran the new validation on the Centipede full distillation. It is costly to run, taking about 5 hours, most of which is the environment simulation. We achieved a reward of $8042 \pm 492$, which is near our previously reported mean of $8083$. The standard deviation is much lower (was 4811) since it is the standard deviation of the mean of each agent rather than the standard deviation over all episodes. Because the means are so close, we do not expect this validation method to provide any shocking changes, but the results will provide stronger evidence of the success of our approach.
> > > > > >
> > > > > > We will continue to run this validation, but note that it will take around 70 hours to validate the rest of the Atari experiments, so all results may not be in by the end of the discussion period. We will continue these experiments regardless.

---

> > > > > > > ### Author Response · Authors · 2024-11-28
> > > > > > >
> > > > > > > We have completed validation with 5 distillations for each MuJoCo environment, and used your proposed validation to redo all the cart-pole and Atari results. We will publish the results with our final paper revision.
> > > > > > >
> > > > > > > None of the results changed drastically, so we have not changed the discussions around the results. We hope that these results satisfy your desire for stronger validation.

---

> > > > > > > > ### Comment · Reviewer_AwMk · 2024-11-28
> > > > > > > >
> > > > > > > > Looks good to me, I have raised my score to 6.

---

> > > > > > > > > ### Author Response · Authors · 2024-12-03
> > > > > > > > >
> > > > > > > > > Thank you for raising the score, and more importantly, thank you for helping us improve our paper. We appreciate your feedback and your engagement throughout the discussion period!

---

> ### Author Response · Authors · 2024-11-15
> **Response to Weakness 1**
>
> W1: We are not certain we fully understand the argument you are making in this point. Our interpretation (of your argument) seems contradictory: you are concerned both that we do not sample enough environment instances, making the distillation too specified; and also that we do sample multiple environment instances, making the distillation too generalized. We also feel that many of these points are levied at reinforcement learning as a whole, which is well out-of-scope for our paper to answer. We will attempt to answer it, but feel free to respond and clarify if we are missing your point.
>
>  Firstly, we consider this argument to be partially based on the semantics of distillation, rather than the results or use-cases. It is true that we frame our distillations as distillations of the environments as a whole, while in truth we sample environment transitions in training. Thus, it is fair to argue that, for example, our distillation on Centipede is a distillation of those 8,000,000 transitions of Centipede seen during training. However, we do not consider this interpretation to affect the quality of our results or the potential use-cases of this method.
>
> The paper you have linked calls into question all RL results using too few iterations of evaluation; we have used results over 1000 environment instances, as well as a clear trend in the training reward curve. We are open to specific suggestions on how to improve these evaluations, yet we consider our models’ consistent high performance from near the end of training throughout evaluation to be indicative of success.
>
> RL environments are not deterministic, but neither is SL training. While this makes evaluation on an RL environment difficult, we make up for this by doing multiple runs on the environment to get our results. We cannot claim, as no PPO agent can ever claim, that our model will always get a certain reward, however, we can claim that it gets an average reward over a given number of runs, which we consider sufficient for RL evaluation. There exists some expected reward over all possible environments that this approximates. We do not see this as a limitation of our model or approach, simply an explainable difference between SL and RL.
>
> While we agree that distilling a single instance of an environment only represents the data of that instance, we do not distill a single instance of the environment. Distillation training was conducted with data gathered from 10 environments run in parallel, over multiple epochs of data gathering: our Centipede distillation thus represents a distillation over 80,000 Centipede environment instances. However, we assert that by training over multiple instances, the distillation-trained models show clear convergence to performance that closely matches that of an RL-trained model. We explicitly chose not to fix the seed of the environments in training or evaluation, thus, our evaluation shows that the distillation works on environments with various seeds that have not been seen in training. This implies that our distiller has learned some general-use knowledge useful for performing the task, just as an RL agent that has been trained over multiple environment instances also learns general-use policies.
>
> It is the case with any finalized distillation that the initialization of the predictive model is the only factor affecting its performance or where it converges to; the rest is deterministic. The distillation does not represent a single environment instance, nor does it intend to; just as dataset distillation does not claim to represent a single shuffling of a training dataset.

---

> > ### Comment · Reviewer_AwMk · 2024-11-24
> >
> > The paper I referenced does not focus on the number of evaluation environments but instead emphasizes training RL agents multiple times using the same hyperparameters while varying the environmental randomness. This approach provides a clearer understanding of the stability and performance of RL algorithms. In contrast, if I understand correctly, your method involves performing distillation once and subsequently training 1,000 agents for evaluation. However, this approach is not equivalent to the one discussed in the referenced paper.
> >
> > In your method, the primary source of randomness in the agents stems from model initialization, as only a single gradient descent step is performed. Moreover, the paper does not appear to present results for distillation across multiple runs. Have you conducted multiple runs of the distillation process? If so, do they produce consistent performance, or do they exhibit variability similar to RL training due to environmental randomness? If not, it is very similar to RL which shows no reason to choose distillation.
> >
> > The paper primarily focuses on distilling a single environment, yet you mention the use of 80,000 different environment instances for the Centipede task. This appears to contradict the claims made in the paper. Could you clarify whether this discrepancy is addressed? For instance, Algorithm 1 suggests the environment is fixed, but this does not align with the stated use of multiple instances. Additionally, what is the reasoning behind using such a large number of environment instances? Why not utilize a single instance or a smaller subset? Have you conducted any ablation studies to investigate this? It is unclear why this critical aspect was not discussed in the paper, given its significant implications.

---

### Official Review · Reviewer_ucbK · 2024-11-03

**Soundness:** 3
**Presentation:** 3
**Contribution:** 3
**Rating:** 8
**Confidence:** 4

**Summary:**

The paper presents an approach that distills a reinforcement learning environment into a synthetic dataset while allowing agent trained under supervised setting with limited resource to reach a comparable performance vs the direct RL training.
It also presents an generalized algorithm to control the difficulty of distillation for estimating the feasibility of the full distillation.

**Strengths:**

Overall the paper is well-written. It provides an clearly-defined algorithm with training graph and pseudocode.
It provides a simple algorithm based on PPO to distill RL environments into a parameterized distiller.
The performance results from an easy task to complex tasks are on par with direct RL training which demonstrates the generalizability and high distillation performance of the algorithm.

**Weaknesses:**

* The experiments do not cover the continuous control problems which are also important part of RL environments. Demonstrating distillation on those tasks can greatly benefits to the community as many robot experiments are under continuous action space.
* If I understand correctly, the final baseline RL agent is determined by the time limit and convergence. But I would image using the same amount of training sample as in the distillation's outer loop for a more fair comparison.
* The cost saving part might be better displayed in a graph. Such as the overall time spend/number of parameter updates vs number of agent trained.

**Questions:**

* In fig 1: What's $D_{\theta}$, same in fig 2.
* Line 197: Why is the gradient destroyed? Isn't the bound non-zero?
* Can you also clarify the instance in sec 4.2, it's the sample generated from the distiller?
* In table 2, why would some experiments did not exceed the random performance while the full distillation did?
* In fig 3, what happens to the sudden increase in (e) subplot?

---

> ### Author Response · Authors · 2024-11-15
>
> Thank you for your review. Here, we’ll address the weaknesses and answer the questions in your review. We have marked anything we will add in the first revision, which we will finish and upload after responding to all reviewers. Some of your concerns will require experimentation (i.e. MuJoCo distillations) and will be added to a revision upon completion.
>
> Please let us know if our response addresses your concerns, or if you have additional concerns, questions, or suggestions.
>
> W1: We agree that the continuous control problems are important. We have begun distilling the MuJoCo environments and will add the results as revisions as they come in. As of now, we have reached 2,000 reward with the Hopper environment.
>
> W2: We agree. We have begun RL training for these baselines and will revise the baselines as these results come in.
>
> W3: We have a simple graph we will add to the appendix for now showing the expected time costs of training $x$ agents using RL vs using distillation (including distillation costs). We will consider more useful visualization methods, such as the ones you’ve suggested here.
>
> Q1: We changed referring to the distilled dataset or distiller from $D_\theta$ to $\\{X\_d, Y\_d\\}\_\theta$, which better reflects our work. Those should read $\\{X\_d, Y\_d\\}\_\theta$, and we will fix that in the revision.
>
> Q2: When clipping occurs, the gradient is 0, as infinitesimal changes to the input will not affect the clipped outcome.
>
> Q3: Yes, “distilled instances” refers to the paired $x,y$ members of the distilled set: $\\{X\_d, Y\_d\\}\_\theta$. The size of the distilled set is a hyperparameter, as the distilled dataset is a parameterized tensor of instances that is learned through the distillation process, rather than generated as output of a distiller network.
>
> Q4: This is partially due to resource constraints on our part: we thought it was more useful to spend those resources on the final distillations rather than on the lower rollback experiments.
>
> Q5: The reward appears to jump more quickly in $l=1$ than other distillations because the $l=1$ distillation is generally the most expensive. We are not certain why distilling all but the first layer is more expensive than full distillation. Perhaps holding the first layer static between distillation-trained networks constrains the networks too much by forcing them to all use one set of baseline features.

---

> > ### Comment · Reviewer_ucbK · 2024-11-23
> >
> > Thank author for addressing my comments and questions.
> >
> > Overall, the paper looks good. I will raise the score once the author address the following questions.
> >
> > * For Q2, is the sentence updated? At line 365, it's still using the old description.
> > * For the distillation size of continuous settings, how is the 64 being decided? And although the equation 3 might not be applied to the continuous setting, it would still be good to see the graph of performance vs ***k*** given the paper want to demonstrate that distillation could reduce the cost of training.
> > * OPTIONAL For Q5: What's `one set of baseline features`? Would be good to have some analysis on the reward jump.

---

> > > ### Author Response · Authors · 2024-11-24
> > >
> > > Thank you for being attentive with us and for being willing to update the scores.
> > >
> > > For the questions you've provided, note that we have limited resources and cannot prioritize too many additional experiments given the review period, but for anything we cannot do in this limited time, we will attempt to complete before the camera-ready version is submitted.
> > >
> > > - We did not update those results yet, but we will submit a new revision with those results tomorrow. We will also add another global response outlining the changes. We ran an experiment training an RL agent on Centipede for 8000 epochs and got a slightly higher reward than before: $8378 \pm 4238$ vs the previous $8160 \pm 4207$. As for the other enviornments, we do not have the computation time to run them for hundreds of thousands of epochs, but since those full distillations did not converge, perhaps it is ok to use the updated results for Centipede only. Given more time, we can continue to run those RL experiments for a more fair comparison on all Atari environments.
> > >
> > > - Due to the time constraints for the review period, we chose an arbitrary k value we assumed would be high enough to work for all the MuJoCo environments. We agree that examining the $k\_{min}$ values would be interesting for continuous action spaces, as well as seeing performance vs k. Given the time constraints, we will begin these experiments on one of the environments and expand them to others as time permits.
> > >
> > > - By "one set of baseline features" we mean that at $l = 1$, only the first layer of the network is trained by RL, while the rest is trained by distillation. Thus, perhaps the single-layer encoding learns features that are difficult to utilize for multiple initializations, or perhaps learning just one layer with RL and the rest with distillation makes the RL learning more difficult. We haven't deeply examined this phenomenon, so that was a rough guess, but we can look at how much that first layer changes throughout learning to see if we can figure out anything useful from it.

---

> > > > ### Comment · Reviewer_ucbK · 2024-11-24
> > > >
> > > > Thank author for the response.
> > > >
> > > > I will update the score.
> > > >
> > > > Please update the Atari and Mujoco (if it's not using the same outer epoch) results in the camera-ready version. Also please update the corresponding sentence with the new experiments. The abstract, intro and discussion do not mention the experiments on the continuous setting yet.
> > > >
> > > > The insight of the first layer change will also be helpful.

---

> > > > > ### Comment · Reviewer_ucbK · 2024-11-26
> > > > >
> > > > > Given the performance of distillation in many environments are not on par with direct RL training and the author claims the distillation is based on the optimal policy. It has to provide some discussion on why is the distillation results are bad. Since the Atari performance is due to convergence, you can provide the analysis on the continuous setting only.
> > > > >
> > > > > Nit: The previous comment said the $D_{\theta}$ will be replaced with $\\{X_d, Y_d\\}_{\theta}$, but the current revision's Fig 1 still uses the old format.

---

> > > > > > ### Author Response · Authors · 2024-11-26
> > > > > >
> > > > > > Thank you for your continued communications. We will continue to run the RL experiments for the same number of outer iterations as the distillation and will put those results into the camera-ready version. We have updated the abstract, intro, and discussion to mention the MuJoCo experiments as well. We will also examine the first layer to see if there are any insights we can glean.
> > > > > >
> > > > > > **Drop in performance**
> > > > > >
> > > > > > Some of these points are already touched on in the paper. We have clarified them in the newest revision as well.
> > > > > >
> > > > > > Note that we do not train on a specific optimal policy; rather, distillation training optimizes a policy similar to standard reinforcement learning.
> > > > > >
> > > > > > Due to resource constraints, we trained each of the MuJoCo distillations for 24 hours parallelized on 4 gpus. We suspect the poorest performance (Inverted Double Pendulum, Walker2D) are due to insufficient time for convergence. As for the others, we note that a drop in performance is expected in distillation. We discuss this a bit in the paper, as we recognize that distillation is a form of lossy compression and we would not expect a one-batch dataset to be able to contain all the complexities of a full environment.
> > > > > >
> > > > > > Dataset Distillation (Wang et al. 2020) demonstrates this drop in performance on supervised learning datasets. They also demonstrate that using a single training initialization reaches higher performance, but overfits on the dataset. In all our experiments, we vary the initialization, as we believe this provides a more generalized and overall more useful distillation, as we demonstrate in our cart-pole k-shot learning experiments. We consider a distillation's generality (i.e. its ability to train multiple model initializations and architectures) to be more useful that higher performance, though both are desirable properties.
> > > > > >
> > > > > > It is also possible that distillation is simply more prone to falling into local minima than standard learning. Since we are optimizing a smaller parameter space than the parameter space of an individual model, the number of "good" policies this space can represent may be less than the number of policies a model can represent. This may lead to more stark minima.
> > > > > >
> > > > > > **Figure 1**
> > > > > >
> > > > > > We apologize for the confusion. We use $T\_\theta$ intentionally in Figure 1.
> > > > > >
> > > > > > Figure 1 is meant to represent task-agnostic distillation. We intentionally use the label "Synthetic Task $T\_\theta$"  to represent an arbitrary parameterized synthetic task. We do not use $\\{X\_d, Y\_d\\}\_\theta$ because that represents distilling to a supervised dataset. While we only distill to supervised datasets in the experiments, we intend Figure 1 to be a broad generalization of distillation.

---

> > > > > > > ### Author Response · Authors · 2024-11-26
> > > > > > >
> > > > > > > We will also extend the MuJoCo experiments we believe may not have converged and add those results to the paper. We do not expect to get those results by the PDF revision deadline, as our computational resources are all busy, but we may report results before the discussion period ends if possible..

---

> > > > > > > > ### Comment · Reviewer_ucbK · 2024-11-27
> > > > > > > >
> > > > > > > > Thanks for updating the paper. My question has been resolved.
> > > > > > > >
> > > > > > > > If time permits, a bigger hyperparameter sweep would be nice. And in the current paper or future paper, something with more exploration involved would be nice to check as the quality of the distillation depends on the RL policy. Such as Random Network Distillation, (based on PPO), or Go-Explore (Ecoffet, Adrien et al.,2019).
> > > > > > > >
> > > > > > > > It would really be appreciated if the author can highlight the change in the revision.

---

> > > > > > > > > ### Author Response · Authors · 2024-11-27
> > > > > > > > >
> > > > > > > > > Thank you for your responses and for increasing the score. We are grateful that we could answer your questions.
> > > > > > > > >
> > > > > > > > > We agree that a better hyperparameter sweep and more exploration could be beneficial, and we will explore these additions as time permits. Thank you for this suggestion.
> > > > > > > > >
> > > > > > > > > We will highlight the salient changes between the first submitted version of the manuscript and the next revision (which we will submit in a few hours once a few more experiments finish). We apologize for not highlighting changes in prior revisions.

---

### Official Review · Reviewer_duG4 · 2024-11-05

**Soundness:** 2
**Presentation:** 1
**Contribution:** 1
**Rating:** 3
**Confidence:** 2

**Summary:**

This paper proposes a PPO-inspired dataset distillation technique.

**Strengths:**

Distillation seems like an interesting technique to reduce the data requirement of reinforcement learning.

**Weaknesses:**

I vote to reject primarily because the motivation for the algorithm and it's empirical evaluation is difficult to follow. I had a difficult time understanding the core takeaways of this paper.

1. Many of the contributions listed can be combined. For instance, contributions 1, 3, 5 and 6 are essentially saying the same thing: this works propose a new distillation technique and demonstrates its effectiveness empirically.
2. Contribution 2 doesn't seem like a contribution; it's simply a task that was created to demonstrate the distillation method. I suggest omitting.
3. "policy gradient methods such as PPO are more sample-efficient, utilizing the entire experience memory rather than randomly sampling from it as in standard DQN learning." This sentence is unclear to me; PPO does not use experience replay, as it is an on-policy algorithm. Also, off-policy algorithms are often more sample efficient than on-policy algorithms, so this statement seems misleading.
4. First paragraph section 3.1: This paragraph motivates building off of PPO, but I think that's all that needs to be said: you build off PPO because it is a reasonably sample efficient on-policy algorithm and is often the go-to algorithm for RL applications.
4. It's unclear how the experiments extend cartpole to N dimensions. A figure for N=2 would make be informative.
4. Table 1 is difficult to parse. What exactly are the core takeaways from this table? Why is it informative to consider different model initializations? the experiments would be easier to understand if hypotheses were stated prior to showing results -- what do we expect to see if the method works, and why?
5. If the method works, it would be useful to understand how it distills the dataset -- which samples are ultimately distilled? Can we glean any insights from it?

**Questions:**

See weaknesses.

---

> ### Author Response · Authors · 2024-11-15
>
> Thank you for your review. Here, we will answer the weaknesses you’ve specified. We have noted where we will add revisions to the paper, though we will wait to post the revised paper after answering all the reviewers’ concerns.
>
> Please let us know if these answer your questions and concerns.
>
>
> W1: While we understand this point, we feel each contribution mentioned represents a distinct point. We agree that contributions 1 and 6 amount to proposing a new technique and demonstrating its effectiveness.
>
> As it stands, we agree with your assessment of contribution 3, but we propose changing it to better reflect a significant point: “Demonstrates k-shot learning on single-batch datasets distilled from ND cart-pole using various initialization distributions and architectures; demonstrating distillation’s generalization to unseen architectures”.  This better highlights a vital contribution supporting one of distillation’s proposed use cases: neural architecture search, which relies on generalization to all architectures in the search space.
>
> Contribution 5 highlights a general-use method for scaling distillation difficulty, which can be used for any distillation method, not only the one we propose.
>
> W2: ND cart-pole provides a flexible and general-use RL task where the action and state space can be expanded without radical changes to the environment. We consider that a contribution worth mentioning and have had prior feedback highlighting ND cart-pole as a unique contribution.
>
> W3: We can clear up this language a bit in the paper. We use “experience memory” to refer to the transitions stored throughout a single epoch of PPO learning, gathered across multiple episodes of play. This may be confused with experience replay, which refers to memory of all prior transitions. Would the term “trajectories” be more appropriate?
>
> We have seen conflicting claims about sample efficiency, so it is likely that there is some confusion in the field in general. We ran small tests on cart-pole, and you are correct, DQN appears to be more sample-efficient. To avoid this controversy, we can remove claims that PPO is more sample efficient. Though as you state in Weakness 4, this does not undermine our decision to use PPO.
>
> W4: Noted: we can simplify that paragraph.
>
> W5: ND cart-pole is explained in Appendix B. We agree that a figure would aid comprehension, and we will add a figure to the appendix.
>
> W6: Table 1 demonstrates evaluations on models sampled from various architecture and initialization distributions, given distillations trained on the distribution $\Lambda$. This has two purposes: 1. To demonstrate that the distillation succeeded by reporting performance by models sampled from $\Lambda$; 2. To explore how the distillation generalizes to unseen initialization distributions and architectures.
>
> Purpose 2 is important because distillations have the potential to overfit---only successfully training models that were sampled from the distribution used in training ($\Lambda$). As stated in the first paragraph of 4.1., our hypothesis matched the current thinking in distillation research: that distillations are very sensitive to overfitting and are not likely to generalize to unseen model distributions. Thus, we expected high performance of models sampled from $\Lambda$ after training on the distillation and low performance on models sampled from other distributions.
>
> As discussed, the table shows that this hypothesis is only partly correct: models sampled from different parameter initialization distributions performed much worse than those from $\Lambda$; however, different architectures performed similarly, and sometimes even better, than those from $\Lambda$. This demonstrates a level of robustness to novel architectures that was not expected. The significance of this is that it helps support calls for using distillation on neural architecture search (a commonly-suggested use-case of distillation), as distillations may be generalizable to the neural architecture search space without using the entire search space in producing the distillation.
>
> W7: The distillation produces synthetic instances that are not required to resemble actual data instances. The instances may provide some limited interpretability, but achieving significant interpretability is out of scope for this paper and an important problem we are tackling as future work. We gained some interpretability from the cart-pole tasks: they show that the action should move the cart towards the direction in which the pole is leaning. However, the distillation of Centipede appears too abstract for any reasonable human interpretation. We will add figures visualizing 1D and 2D cart-pole in the appendix.

---

> ### Author Response · Authors · 2024-11-26
> **Request for reviewer discussion**
>
> Dear reviewer,
>
> We have not heard from you after our rebuttal over a week ago, and the original discussion period is coming to an end. Due to your low scores and low confidence, we have looked forward to your response to our rebuttal and the improvements we have made given your feedback. If we are to improve the paper to better respond to any remaining criticisms, we would need to hear back soon, as the deadline for PDF resubmissions is nearing.
>
> As we have shown with our continued and timely responses with the other reviewers, we are willing to continue dialogue to improve our paper and the reviewers' understanding of our work. We ask you to please examine our rebuttal and to take into account the extensive improvements we have made on our paper throughout this past week. We believe we have addressed all the concerns in your review adequately, though we disagreed on the proposed changes to the contributions. If you have additional issues, or feel we did not adequately address any criticisms, please let us know so that we can address them.
>
> As your review's scores diverge most from the other reviews, we consider dialogue with you to be a priority. In addition, the low confidence you placed in your review indicates that there is much room for us to clarify our positions and our methods, both in the paper and in dialogue. Given adequate time, we believe we can respond to all your concerns. Luckily, the discussion period has been extended, and we will endeavor to continue to be available throughout the extended period.

---

### Author Response · Authors · 2024-11-19
**Paper Revisions**

In response to reviewer feedback, we have made the following changes in the latest revision:

- Performed experiments with continuous MuJoCo environments and added results to paper
- Clarified contribution 3: "Demonstrates k-shot learning on single-batch datasets distilled from ND cart-pole using various initialization distributions and architectures; demonstrating distillation’s generalization to unseen architectures"
- Changed use of "experiences," "memory," and "experience memory" to "observation sequences," "transitions," or "RL trajectories" where appropriate for more accurate description of PPO learning.
- Revise 3.1. for clarity, removed most of 1st paragraph & all references to PPO being sample-efficient
- Added figure of ND cart-pole and visualizations of 1D and 2D cart-pole distillations in appendix
- Added visualization of cost saving over number of models produced in appendix
- Removed remaining references to distilled dataset as $D\_\theta$
- Clarified evaluation procedure
- Moved Figure 2 (distillation algorithm figure) to appendix

---

> ### Author Response · Authors · 2024-11-25
> **Paper Revision 2**
>
> Thank you all for your feedback. We have implemented the following revisions to our paper in accordance with the feedback and with additional experiments suggested by the reviewers:
>
> - Added performance from uniform random agent for MuJoCo to contextualize distillation's improvements
>
> - Ran Centipede RL for same number of steps as Centipede distill for fairer comparison
>
> - Clarified that we are not distilling the entirety of environments, rather we are distilling optimal policies.
>
> - Included MuJoCo experiments in abstract, intro, and discussion
>
> - Updated Hopper distillation results: reporting results over 5 distillations. Experiments w/ other MuJoCo envs are in progress.
>
> - Results for using diffusion model as baseline added to appendix w/ discussion
>
> - Line number references w/ algorithm
>
> - Fixed minor errors w/ formatting (page headers show up correctly)

---

> > ### Author Response · Authors · 2024-11-26
> > **Paper Revision 3**
> >
> > We are hoping to post one more revision with improved validation results for Atari and MuJoCo experiments. After that, we will continue to revise our version of the manuscript with reviewer feedback and experiment results. We will discuss these in individual reviewer comments rather than global comments, unless we feel it is a concern to all reviewers.
> >
> > We have implemented the following revisions:
> >
> > - Updated k-shot cartpole experiments (tested 100 agents over 100 episodes each), and updated description. The numerical results are largely similar, so we have not provided additional commentary, but we believe this validation method strengthens the validity of our results.
> >
> > - Added random performance for cart-pole and Atari experiments in appendix for clearer results comparisons.
> >
> > - Clarified reasons for distillation's lower performance than direct-task learning.

---

> > > ### Author Response · Authors · 2024-11-28
> > > **Final Paper Revision**
> > >
> > > Here are the changes for the final paper revision:
> > >
> > > - Added blue text highlighting to signify changes made since the initial submission (we apologize if minor changes have been missed). The blue highlighting will be removed for the camera-ready version. Note that the table values were not highlighted, though using the improved validation techniques suggested by the reviewers, we have updated:
> > >   - Table 1: all values
> > >   - Table 2: all distillation values, Centipede RL
> > >   - Table 3: added entire table
> > >
> > > - Used improved validation for Centipede and MuJoCo (all distillation values updated)

---

### Meta-Review · Area_Chair_xYxA · 2024-12-23

**Metareview:**

This paper develops an approach to distill the transitions sampled by PPO, to use them to initialize a new policy on a later training run. The paper implements this idea using a meta-learning-variant of PPO.

There was substantial discussion on this paper and a number of points were raised by the reviewers. The most important ones revolved around (i) evaluation methodology, (ii) subpar performance to training from scratch, (iii) motivation and (iv) baselines. The authors have satisfactorily addressed the first concern, but the others remain. This paper is poorly written, both the narrative and the mathematical rigor of the paper could be improved substantially.

This approach is quite ill-motivated, e.g., there is no generalization to new tasks or morphologies, indeed if one was able to run on the environment once, there is no more need to distill data from it. The idea of dataset distillation is about using this data for addressing new environments. The second concern is more major. There are a number of existing ideas in the RL literature, e.g., imitation learning on subsampled data, and offline RL methods in general, that could be thought of as appropriate baselines for this work. The paper has not considered any of these existing ideas. If one thinks formally about the setup in this paper, the distilled dataset is indeed the states and actions of a near-optimal PPO policy.

**Additional Comments On Reviewer Discussion:**

Reviewer duG4 had some minor clarification and stylistic questions about the paper. The authors have addressed these questions carefully in the rebuttal, but the reviewer did not update their score. I am therefore going to discount this review substantially.

Reviewer ucbK had questions about applicability to continuous control problems and concerns as to why the performance of the distillation-based approach was always sub-par compared to training using reinforcement learning. There was a lot of rather speculative discussion in this discourse with this reviewer (by both the reviewer and the authors).

Reviewer AwMk pointed out similar work on dataset distillation for offline RL. They also had concerns about the motivations of this work and the quality of writing (which I share). There was a lot of back and forth with the authors on the exact evaluation procedure (roughly, how many agents are created using distillation and how many episodes are used to average the performance). After this discourse, the authors have changed the reported numbers in the paper and the reviewer is onboard with these new experiments.

Reviewer P1g3 was concerned about missing baselines and motivation. I agree with both these points. The reviewer is also concerned about the approach in general, because even though the authors are calling this approach “dataset distillation”, it is actually a distillation of the transitions of a near-optimal PPO policy. As such, behavior cloning/imitation learning on subsampled data or an offline RL algorithm would be perfect baselines for the proposed method. The authors have not provided any comparisons with baselines. Surprisingly, the reviewer has pointed out a lot of gaps in the paper and engaged in a robust discussion. But they have given a very high score of 8/10, so I will calibrate their score to the trends in the discussion, and argue that this is more a score of 5 or 6/10.

---

### Decision · Program_Chairs · 2025-01-22

Reject